# PROMPT REINFORCING FOR LONG-TERM PLANNING OF LARGE LANGUAGE MODELS

## ABSTRACT

Large language models (LLMs) have achieved remarkable success in a wide range of natural language processing tasks and can be adapted through prompting. However, they remain suboptimal in multi-turn interactions, often relying on incorrect early assumptions and failing to track user goals over time, which makes such tasks particularly challenging. Prior works in dialogue systems have shown that long-term planning is essential for handling interactive tasks. In this work, we propose a prompt optimisation framework inspired by reinforcement learning, which enables such planning to take place by only modifying the task instruction prompt of the LLM-based agent. By generating turn-by-turn feedback and leveraging experience replay for prompt rewriting, our proposed method shows significant improvement in multi-turn tasks such as text-to-SQL and task-oriented dialogue. Moreover, it generalises across different LLM-based agents and can leverage diverse LLMs as meta-prompting agents. This warrants future research in reinforcement learning-inspired parameter-free optimisation methods.

## 1 INTRODUCTION

Large language models (LLMs) have shown an extraordinary ability to perform a wide range of tasks, from generating images in various styles to writing code in different programming languages for diverse purposes. LLMs are typically post-trained using reinforcement learning from human feedback (RLHF) (Ouyang et al., 2022), where they receive single-turn rewards for individual responses rather than rewards reflecting the quality of an entire multi-turn conversation. This limits their effectiveness in interactions where tasks are underspecified and clarified over time, often leading to early mistakes, incorrect assumptions, and cascading failures (Laban et al., 2025). On the other hand, prior work in dialogue systems demonstrates that long-term planning is vital for interactive tasks, making it essential for LLMs (Young, 2002; Young et al., 2013).

Directly optimising LLMs could improve their ability to plan across multiple turns, e.g., supervised fine-tuning with low-rank adaptation (Hu et al., 2022), direct preference optimisation (Feng et al., 2025b), continuous prompting (Lester et al., 2021; Qin & Eisner, 2021; Li & Liang, 2021; Liu et al., 2023), or reinforcement learning with dialogue-level rewards (Feng et al., 2025a); however, these approaches are often impractical for real-time updates due to high computational costs, especially with limited local resources, and are incompatible with API-only LLMs.

Gradient-free methods, such as instruction-feedback-refine pipelines (Peng et al., 2023; Shinn et al., 2023; Yao et al., 2023; Elizabeth et al., 2025), avoid parameter updates but rely on frequent API calls during inference, leading to inefficiency. Meta-prompting and existing prompt optimisation techniques focus on input-output learning without explicitly modelling long-term planning (Yang et al., 2024a; Tang et al., 2025; Pryzant et al., 2023; Yuksekgonul et al., 2025).

To address these limitations, we propose Reinforced Prompt Optimisation (RPO). The structure of RPO is shown in Figure 1. This meta-prompting approach enhances the long-term planning ability of LLMs by iteratively refining an initial prompt based on natural language feedback, where the initial prompt can be crafted by experts or generated from a corpus via meta-prompting (Zhou et al., 2023; Pryzant et al., 2023; Ye et al., 2024).

In RPO, an LLM-based system interacts with an environment, such as real or simulated users, in tasks like information seeking or medical QA. A feedbacker, either a human or an LLM, provides

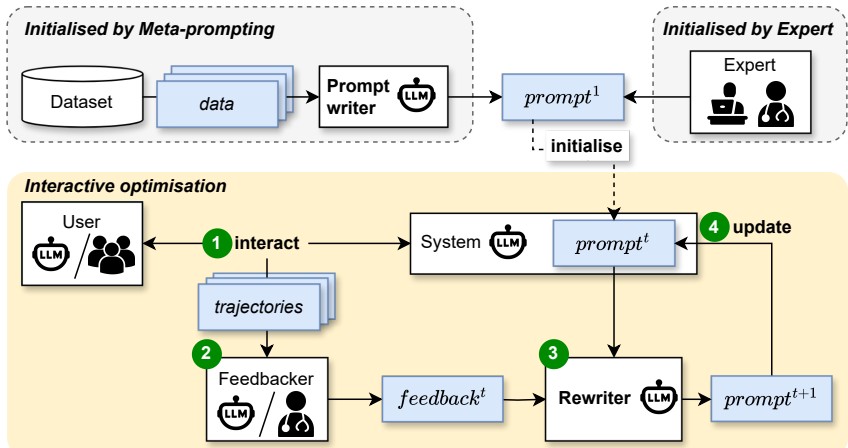

Figure 1: The structure of Reinforced Prompt Optimisation (RPO). The initial $prompt^1$ can be generated by LLMs or written by experts. In interactive optimisation, the system will first interact with the environment, e.g., simulated or real users. The feedbacker, e.g., human experts or LLMs, will provide textual feedback based on trajectories. The rewriter generates a new prompt based on the original prompt and the textual feedback to update the system's original prompt. One cycle of interactive optimisation is called an epoch, and we use *superscripts* to denote the epoch number.

turn-level textual feedback inspired by temporal difference (TD) error. As shown in the right part of Figure 2, for each turn $t_i$, the LLM-generated feedback includes: (1) predicted user emotion in the next turn elicited by the system response $a_i$, (2) a forecast of dialogue success or failure, and (3) suggestions based on the subdialogue $t_{1:i}$. These are then aggregated into dialogue-level feedback.

A separate LLM-based rewriter refines the prompt based on the feedback and the previous prompt. Experience replay is applied by leveraging feedback–prompt pairs from both the current and past iterations. The updated prompt is used in future interactions. More details can be found in Section 3. Inspired by these well-studied reinforcement learning concepts, the goal of RPO is to effectively strengthen the system agent's long-term planning ability and overall task success.

Our contributions are as follows:

- We propose Reinforced Prompt Optimisation (RPO), a meta-prompting framework that improves LLMs' long-term planning in multi-turn tasks by iteratively updating prompts based on natural language feedback.

- We explore leveraging the concept of temporal difference (TD) error in the LLM-based feedback generation and experience replay in rewriting, enabling efficient and lower-variance prompt optimisation.

- Our method can leverage external expert reward signals without revealing the prompt of the LLM-based system and is flexible with respect to the choice of LLM backbones for the system or meta-prompting agent.

## 2 RELATED WORK

**Gradient-based optimisation for LLMs** For high parameter counts, training or fine-tuning an entire large language model is infeasible since it requires a huge amount of computational resources. As a result, parameter-efficient fine-tuning, such as training only part of the model or freezing the model and training an adapter, is widely used to refine LLMs (Hu et al., 2022; 2023; Lialin et al., 2023). On the other hand, continuous prompting, e.g., prefix-tuning and soft-prompting, is also popular to adapt LLMs to specific tasks or improve their performance (Lester et al., 2021; Qin & Eisner, 2021; Li & Liang, 2021; Liu et al., 2023). By updating inputs of every attention layer (Li & Liang, 2021), or task-related vectors (Lester et al., 2021), these methods can achieve comparable performance to full fine-tuning across various model sizes and tasks (Liu et al., 2022). Although

these methods can improve LLMs effectively, they do not apply to API-access-only LLMs, and such training processes cannot be carried out in real-time.

**Self-feedback** To improve the performance of text-based prompts, various prompting styles are proposed, e.g., Chain-of-Thought (Wei et al., 2022) or ReAct (Yao et al., 2023). These prompting methods encourage LLMs to reason before taking action or generating responses, which leads to better performance. However, optimising the prompt for better performance by manual trial and error is inefficient. Instead, self-feedback methods are introduced to refine the LLMs' response, e.g., LLM-augmenter generates feedback by itself and leverages external knowledge to rewrite its response (Peng et al., 2023), and Reflexion summarises previous interactions with the environment as 'reflections' to improve the model's response (Shinn et al., 2023; Madaan et al., 2023).

While this demonstrates the ability of LLMs for self-correction, these self-feedback methods rely on frequent API calls since their original prompt is not optimal. As a result, the computation cost and latency during inference are not negligible.

**Prompt optimisation** Meta-prompting methods are widely used to generate a prompt without human editing. The automatic prompt engineer (APE) method leverages an LLM, which is instructed to generate an initial prompt and selects the prompt with the best performance on the target task (Zhou et al., 2023). Automatic prompt optimisation (APO) further employs a self-feedback module to provide textual feedback, which gives suggestions on how to edit the old prompt (Pryzant et al., 2023). Ye et al. (2024) propose a meta-prompt LLM to edit the original prompt step-by-step. Kong et al. (2024) and Cheng et al. (2024) train a sequence-to-sequence model for prompt rewriting by reinforcement learning and preference data, respectively. Yang et al. (2024a) propose optimisation by prompting (OPRO), which leverages LLMs to rewrite the original prompt based on a corresponding performance score. To leverage experience, Zhang et al. (2023) model LLMs as semi-parametric RL agents with memory storing task data, actions, and $Q$-value estimates for few-shot in-context learning. Zhang et al. (2024) propose Agent-Pro, which constructs policy-level reflections according to the numerical feedback from the environment and improves its policy incrementally. Tang et al. (2025) introduce the Gradient-inspired LLM-based Prompt Optimizer (GPO), which updates the prompt iteratively based on numerical feedback and controls the edit distance through a cosine-based decay strategy. TextGrad generates textual feedback based on the user input and system output for prompt rewriting (Yuksekgonul et al., 2025). Although these methods demonstrate promising performance in generating or improving prompts, they focus on single-turn tasks. Our approach addresses multi-turn interactions, where prompts are updated with temporally grounded feedback to enhance long-term planning ability.

**Learning ability of LLMs via prompting** Although transformers are universal approximators (Yun et al., 2020) and in-context learning in LLMs can be viewed as implicit fine-tuning (Dai et al., 2023), the following remain open questions: Can we prompt LLMs for arbitrary tasks, and what are the limitations of in-context learning?

Petrov et al. (2024) highlight the limitations of context-based fine-tuning methods, e.g., in-context learning, prompting, and prefix tuning, for new task learning in transformers. Specifically, transformers struggle to acquire new tasks solely through prompting, as prompts cannot change the model's attention patterns. Instead, they can only bias the output of the attention layers in a fixed direction and elicit skills learned through pre-training. In other words, only models with billions of parameters trained on vast, diverse datasets are capable of in-context learning, adapting to new tasks through examples or instructions without modifying their underlying weights. Therefore, we focus on fundamental models large enough to demonstrate their in-context learning ability, to investigate reinforcement prompt optimisation, which is fully composed of in-context learning with LLMs.

## 3 REINFORCED PROMPT OPTIMISATION

Inspired by the gradient-based optimisation and reinforcement learning algorithms, where a model is initialised from pretraining and then further updated by on-policy learning based on interactions with the environment, we propose the Reinforced Prompt Optimisation (RPO) method (as shown in Figure 1 and the pseudo code can be found in Algorithm 1). The initial instruction can be generated

by a prompt writer $\text{LLM}_P$ such as the automatic prompt engineer (APE) (Zhou et al., 2023) (the upper left part of Figure 1) or written by human experts (the upper right part of Figure 1).

In the interactive optimisation (the lower part of Figure 1), the **system** will interact with the environment, e.g., human users or simulated users, and generate several multi-turn *trajectories*, which, for example, can be task-oriented dialogue or medical question-answering. Then the **feedbacker**, which can be a language model $\text{LLM}_F$ or human experts, will provide textual feedback to guide the optimisation direction for the **rewriter** $\text{LLM}_R$, which will generate a new prompt to improve the system's performance based on the feedback and original prompt.

We emphasise that although our method shares a feedback–rewrite structure similar to self-refine approaches, the key difference lies in the target of refinement. Self-refine methods polish the agent's output, whereas our method updates its instruction. In other words, we treat the system's instruction as a textual parameter to be modified, which reduces serving costs and latency by lessening the need for a multi-agent-style feedback and rewriting pipeline.

## 3.1 FEEDBACK GENERATION

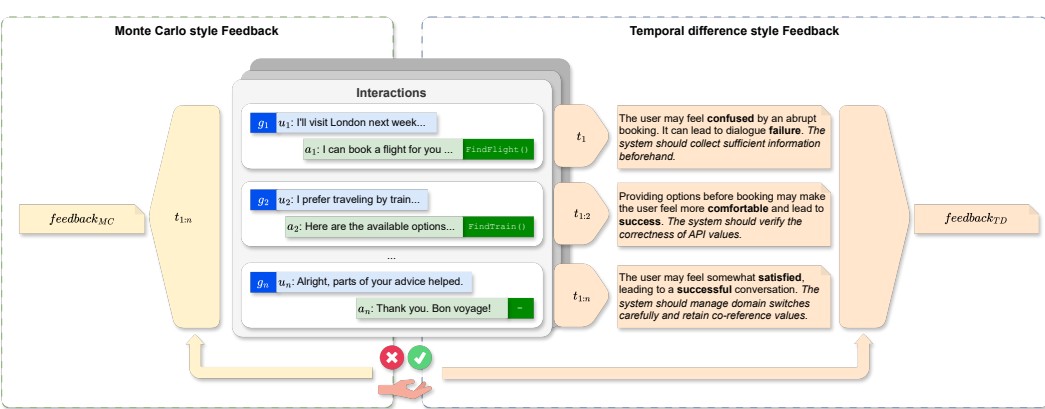

Figure 2: Workflow of feedback generation by an LLM. The Monte Carlo–style feedback (left) is generated after the entire interaction is completed, whereas the Temporal Difference–style feedback (right) consists of turn-level sub-feedback. Each sub-feedback includes a prediction of next-turn user satisfaction, a prediction of goal success, and an actionable suggestion.

As shown in Figure 2, we consider two approaches for generating feedback via LLMs: Monte Carlo (MC)-style and Temporal Difference (TD)-style feedback generation.

The **MC-style feedback** is produced only after the entire dialogue trajectory $(t_{1:n})$ has been completed (the prompt of the MC-style feedbacker is shown in Figure 17):

$$feedback_{\text{MC}} = \text{LLM}_F(t_{1:n}) \tag{1}$$

This approach is commonly used in single-turn tasks such as sequence classification, named-entity recognition, or one-turn question answering (Pryzant et al., 2023; Ye et al., 2024; Wang et al., 2024; Tang et al., 2025; Yuksekgonul et al., 2025). It typically yields prompt modification suggestions based on a global success or failure signal. While this captures the overall quality of the interaction, it collapses the inherently multi-turn nature of real-world interactions into a single outcome.

In contrast, our proposed **TD-style feedback** incorporates turn-level evaluations:

$$feedback_{\text{TD},j} = \text{LLM}_F(t_1, feedback_{\text{TD},1}, t_2, feedback_{\text{TD},2}, \dots, t_j), \tag{2}$$

where $feedback_{\text{TD},j}$ is the turn-level feedback at turn $j$. All turn-level feedback, $feedback_{\text{TD},1:j}$, will be summarised by $\text{LLM}_F$ into a final $feedback_{\text{TD}}$ afterwards (details of the prompt are shown in Figure 18, and examples of turn-level, dialogue-level and final feedback are shown in Figure 8, Figure 9 and Figure 10, respectively). Rather than waiting until the dialogue ends, the feedbacker provides incremental assessments at each turn, including the prediction of user sentiment and expected dialogue success, along with actionable suggestions.

In other words, TD-style feedback treats the immediate user response as a short-term reward (Ghazarian et al., 2022), while also estimating long-term outcomes such as task success. This idea can be formalised through the *TD error*, which balances short-term reward and long-term estimation:

$$\delta_t = r_t + \gamma V(s_{t+1}) - V(s_t) \tag{3}$$

where $r_t$ corresponds to the short-term reward (e.g., user sentiment after the current turn), $V(s_t)$ is approximated by the previous turn-level feedback, and $V(s_{t+1})$ represents the estimated long-term value of continuing the dialogue toward successful task completion[1]. This dual perspective enables the system to refine both local decision-making at the turn level and global trajectory planning across the full interaction.

## 3.2 APPLYING FEEDBACK TO THE PROMPT

Unlike gradient-based optimisation, where gradients can be added or subtracted from model parameters, incorporating textual feedback into prompts is non-trivial. One cannot concatenate or remove arbitrary text from the original prompt without risking incoherence or loss of functionality. To address this, we introduce a basic *rewriter* $\text{LLM}_R$ to apply textual feedback on the original prompt:

$$prompt^{i+1} = \text{LLM}_R(prompt^i, feedback^i), \tag{4}$$

where $i$ denotes the epoch index. Its instruction is shown in Figure 15.

Inspired by experience **replay** in reinforcement learning (Andrychowicz et al., 2017), the rewriter can leverage not only the prompt and feedback from the current epoch, but also those from previous epochs (its instruction is shown in Figure 16):

$$prompt^{i+1} = \text{LLM}_R(prompt^i, feedback^i, prompt^{i-1}, feedback^{i-1}, \ldots, prompt^1, feedback^1). \tag{5}$$

Reinforced Prompt Optimisation (RPO) alleviates the need for task-specific manual prompt engineering by automating prompt creation and refinement entirely through LLMs. The feedback signal may originate from either simulated environments or human users. Importantly, while the feedbacker and rewriter themselves are LLMs that require prompts, these prompts are task-independent and need to be specified only once. Optimising the prompts of these meta-prompting agents lies beyond the scope of this work and is left for future research.

## 4 EXPERIMENT SETTINGS

In this study, we focus on iterative meta-prompting by leveraging textual feedback from the environment. We conduct experiments on three challenging human–machine interaction tasks that require multiple turns: Text-to-SQL, Task-oriented Dialogue, and Medical Question-answering (Section 4.1). An overview is shown in Figure 3. Our meta-prompting components are *task-agnostic* (Section 4.2). They are designed to optimise the prompt of interactive LLM-based systems (Section 4.3). Furthermore, to assess how different prompts affect system performance, all prompts are in a zero-shot in-context learning fashion[2], consisting only of task descriptions without examples.

### 4.1 TASKS

**Text-to-SQL**  Laban et al. (2025) proposed 6 tasks to study the performance drop of LLMs from fully-specified user queries to multi-turn interactions. The multi-turn, sharded instruction (e.g., Shard 1 conveys the high-level intent, and subsequent shards provide incremental clarifications) is partitioned based on the single-turn, fully-specified instruction from the original dataset. The largest decline occurs in the Text-to-SQL task, which we therefore select to study under different prompt optimisation methods, using instructions and databases from the Spider dataset (Yu et al., 2018).

In this task, the system agent receives a database schema at the start of the interaction and generates SQL queries from user queries in natural language. We evaluate both closed-source LLMs (GPT-4o

---

[1] If the task is episodic, the discount factor $\gamma$ can be viewed as set to one (Sutton et al., 2018), so there is no need to specifically take care of it in text.

[2] Following Brown et al. (2020), this is in-context learning since task descriptions are given as context, but also zero-shot because no demonstrations are included.

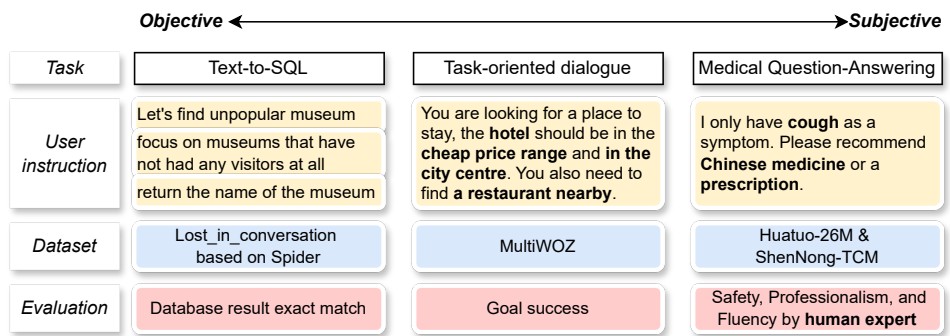

Figure 3: The summary of our experiment tasks.

mini, Gemini-2.0-flash) and open-source LLMs (Llama-3.1-8B, Llama-3.1-70B, Llama-4-scout) to test whether prompt optimisation generalises across different LLMs. The agent is optimised in the multi-sharded environment and evaluated by *functional accuracy*, requiring generated SQL queries to exactly match the reference outputs across all databases.

**Task-oriented Dialogue**  To evaluate on a more realistic scenario, we conduct experiments on MultiWOZ 2.1 (Budzianowski et al., 2018; Eric et al., 2020), containing 10k human-to-human conversations on information-seeking, recommendations, and reservations across multiple domains. In this work, we focus on the attraction, hotel, restaurant, and train domains, under the ConvLab-3 framework (Zhu et al., 2023). Each user goal of the simulated user is a plain-text description of requirements, e.g., *"You are looking for a place to stay, the hotel should be in the cheap price range and in the city centre. You also need to find a restaurant nearby."*

The system agent is FnCTOD (Li et al., 2024), built with GPT-4o mini. In comparison to the standard, single-stage LLM-based system, FnCTOD consists of two parts: dialogue state tracking as a function call to access external databases, and response generation based on function call results. Both prompts are subject to optimisation. The performance of the system is measured by *success rate*, i.e., whether the recommended entities satisfy user goals and all the requested information is fulfilled, based on a rule-based evaluator in ConvLab-3.

**Medical Question-Answering**  To evaluate our system in a more human-centred setting and how well prompting can improve the model's performance in a domain that is not common in the pre-training data, we use two medical question-answering datasets: Huatuo-26M (Wang et al., 2025) and ShenNong-TCM (Wei Zhu & Wang, 2023). The questions in Huatuo-26M and ShenNong-TCM are collected from the internet, e.g., encyclopedias, books, literature, and web corpus, or generated by an LLM based on a traditional Chinese medicine entity graph in Huatuo-26M and ShenNong-TCM, respectively. Simulated users act based on descriptions in plain text, related to general medicine or traditional Chinese medicine, e.g., *"我只有咳嗽這一個症狀，請幫我推薦中藥或者方劑。(I only have cough as a symptom. Please recommend Chinese medicine or a prescription.)"*.

The system agent is built with GPT-4o mini, interacting with users in single-turn or multi-turn settings. It does not access external knowledge bases but relies solely on pre-training knowledge. At each epoch, an expert with degrees in general medicine and traditional Chinese medicine provides feedback on 10 interactions. For evaluation, three different experts compare 2 systems on 30 interaction pairs in general medicine and 30 in traditional Chinese medicine per expert (90 per domain in total), based on safety, professionalism, and fluency, following the setting in Yang et al. (2024b).

### 4.2 META-PROMPTING COMPONENTS

In the interactive optimisation phase, the feedbacker $\text{LLM}_F$ and rewriter $\text{LLM}_R$ are built with closed-source LLMs, e.g. GPT-4o mini (OpenAI et al., 2024) and Gemini-2.0-flash (Gemini Team et al., 2024), or open-source LLMs, e.g. Llama-3.1-8B, Llama-3.1-70B (Grattafiori et al., 2024), and Llama-4-scout (MetaAI, 2025). More detail is shown in Table 3. Across different tasks, the prompts of $\text{LLM}_F$ and $\text{LLM}_R$ remain fixed, highlighting the task-independent role of these components.

### 4.3 Optimisation and Evaluation

We start by collecting interactions using the initial prompt and user instructions sampled from the training set. The feedbacker receives 10 interactions, since the context length of the LLM-based feedbacker is limited, and to efficiently incorporate human expert feedback. At each epoch, the rewriter generates 2 new prompts based on the previous prompt and the feedback. New interactions are collected with each candidate prompt, and the one with the highest score on the validation set (based on automatic metrics or human experts, depending on the task) is chosen for the next iteration. The number of training epochs is a manually chosen hyperparameter: we use 5 for Text-to-SQL, 8 for Task-oriented Dialogue, and 3 for Medical Question-Answering.

**Baselines** In our experiments, we compare various prompt optimisation methods. *Black-Box Prompt Optimization (BPO)* fine-tunes Llama-2-7B-Chat (Touvron et al., 2023) for prompt rewriting based on preference learning (Cheng et al., 2024). *Automatic Prompt Optimisation (APO)* uses the user input, system output, and label to generate feedback (Pryzant et al., 2023). For multi-turn interactions, golden labels are infeasible since multiple solution paths exist; thus, we use a binary success/failure label. *Gradient-inspired Prompt Optimizer (GPO)* iteratively updates prompts using numerical feedback, e.g., functional accuracy for Text-to-SQL, task success for dialogue (Tang et al., 2025). *MC-style (TextGrad)* (Yuksekgonul et al., 2025) processes the entire conversation and generates textual feedback, as mentioned in Section 3.1.

## 5 Results and Discussion

### 5.1 Robustness and Generalisability

Table 1: Functional accuracy with a 95% confidence interval of Text-to-SQL system agents built on five LLMs optimised with various methods. Oracle$_{\text{Full}}$: An oracle baseline in a single-turn setting with fully-specified user queries. The final two columns show the average score (Mean) and the relative improvement ($\Delta\%$) over the Baseline$_{\text{Sharded}}$. **Bold** scores are significantly better than the others ($p < 0.05$).

| Method | LLM of the *system* agent | | | | | Mean | $\Delta\%$ |
|---|---|---|---|---|---|---|---|
| | GPT | Gemini | Llama-4 | Llama-8B | Llama-70B | | |
| Baseline$_{\text{Sharded}}$ | $0.402_{\pm 0.002}$ | $0.514_{\pm 0.027}$ | $0.206_{\pm 0.013}$ | $0.224_{\pm 0.016}$ | $0.318_{\pm 0.014}$ | 0.333 | - |
| BPO | $0.318_{\pm 0.061}$ | $0.397_{\pm 0.069}$ | $0.220_{\pm 0.026}$ | $0.185_{\pm 0.028}$ | $0.241_{\pm 0.061}$ | 0.272 | $-15.7$ |
| APO | $0.374_{\pm 0.009}$ | $0.523_{\pm 0.008}$ | $0.318_{\pm 0.012}$ | $0.290_{\pm 0.013}$ | $0.336_{\pm 0.004}$ | 0.368 | 16.9 |
| GPO | $0.458_{\pm 0.008}$ | $0.523_{\pm 0.016}$ | $0.299_{\pm 0.017}$ | $0.290_{\pm 0.015}$ | $0.308_{\pm 0.011}$ | 0.376 | 17.5 |
| MC-style | $0.459_{\pm 0.018}$ | $0.551_{\pm 0.015}$ | $0.250_{\pm 0.013}$ | $0.346_{\pm 0.021}$ | $0.332_{\pm 0.007}$ | 0.388 | 20.4 |
| RPO$_{\text{TD}}$ (ours) | $0.439_{\pm 0.011}$ | $0.561_{\pm 0.013}$ | $0.336_{\pm 0.014}$ | $0.318_{\pm 0.009}$ | $0.383_{\pm 0.011}$ | 0.408 | 28.9 |
| RPO$_{\text{TD+replay}}$ (ours) | $\mathbf{0.528}_{\pm 0.011}$ | $\mathbf{0.607}_{\pm 0.018}$ | $\mathbf{0.383}_{\pm 0.013}$ | $\mathbf{0.467}_{\pm 0.022}$ | $\mathbf{0.402}_{\pm 0.012}$ | **0.477** | **54.2** |
| Oracle$_{\text{Full}}$ | $0.893_{\pm 0.007}$ | $0.841_{\pm 0.007}$ | $0.729_{\pm 0.017}$ | $0.505_{\pm 0.017}$ | $0.748_{\pm 0.014}$ | 0.743 | 140.2 |

**System agents as different LLMs** Table 1 shows the results of optimising system agents built on five LLM backbones for the text-to-SQL task. Prompt optimisation methods aim to improve system agents in the multi-sharded setting, i.e., the user only reveals part of the information in one turn. For comparison, Oracle$_{\text{Full}}$, a single-turn setting where the user query is fully specified at once, is taken as an upper bound. The performance gap between Baseline$_{\text{Sharded}}$ and Oracle$_{\text{Full}}$ (average $0.333$ vs. $0.743$) highlights the difficulty LLMs face in handling multi-turn interactive tasks.

The prompts optimised by BPO do not improve performance; in some cases, they even degrade it, especially when applied to systems built on different model families, e.g. GPT-4o mini or Gemini-2.0-flash. It is not surprising that BPO is inferior since it is optimised for revising prompts for single-turn tasks and does not generalise well to multi-turn scenarios. Its generalisability is further limited by its dependence on the Llama family as the backend, resulting in suggestions that may not transfer to GPT- or Gemini-based system agents. On the other hand, RPO$_{\text{TD}}$ outperforms prior approaches when the system agent is built with Gemini-2.0-flash, Llama-4-scout, and Llama-3.1-70B. In contrast, RPO$_{\text{TD+replay}}$ achieves the best overall performance, with an average score of

0.477 (+54.2% over Baseline_{Sharded}). Llama-3.1-8B benefits the most, since its performance optimised by RPO_{TD+replay} (0.467) nearly matches the oracle fully-specified setting (0.505). The consistent improvements across closed-source (GPT-4o-mini, Gemini-2.0-flash) and open-source (Llama variants) models demonstrate the robustness of our approach and the effectiveness of combining temporal-difference style feedback with replay.

However, despite substantial gains over the sharded baseline, a gap to the baseline with the fully-specified user query (average 0.477 vs. 0.743) underscores that prompt optimisation can mitigate, but not fully eliminate, the degradation caused by multi-turn interactions.

**Prompt optimisation with different LLMs** Table 2 reports the success rates of FnCTOD (Li et al., 2024) when optimised by different prompt optimisation methods across five LLM backbones. The baseline system achieves a success rate of 0.420, while all optimisation methods substantially improve performance. Among prior approaches, MC-style feedback yields the strongest results with a mean success rate of 0.565 (+34.4% over baseline), slightly outperforming APO and GPO. Our proposed methods consistently surpass these baselines. In particular, RPO_{TD} achieves a mean score of 0.575 (+37.0%), demonstrating the advantage of trajectory-driven optimisation. When combined with the rewriter with experience replay, RPO_{TD+replay} delivers the best performance across all LLMs, reaching an average success rate of 0.619, corresponding to a relative improvement of 47.3%. The gains are consistent across all five LLMs, confirming that our approach is robust and generalisable, independent of the underlying model of the meta-prompting agents.

Table 2: The success rate with a 95% confidence interval of the task-oriented dialogue system, FnCTOD (Li et al., 2024), improved by various prompt optimisation methods leveraging 5 different LLMs. The initial success rate of FnCTOD is 0.420. **Bold** scores are significantly better than the others ($p < 0.05$).

| Method | LLM of the *meta-prompting* agent | | | | | **Mean** | $\Delta\%$ |
|---|---|---|---|---|---|---|---|
| | GPT | Gemini | Llama-4 | Llama-8B | Llama-70B | | |
| APO | $0.540_{\pm 0.030}$ | $0.560_{\pm 0.018}$ | $0.540_{\pm 0.018}$ | $0.560_{\pm 0.018}$ | $0.560_{\pm 0.018}$ | 0.552 | 31.4 |
| GPO | $0.579_{\pm 0.021}$ | $0.541_{\pm 0.008}$ | $0.571_{\pm 0.046}$ | $0.554_{\pm 0.017}$ | $0.526_{\pm 0.053}$ | 0.554 | 32.0 |
| MC-style | $0.567_{\pm 0.042}$ | $0.549_{\pm 0.010}$ | $0.575_{\pm 0.038}$ | $0.560_{\pm 0.029}$ | $0.572_{\pm 0.044}$ | 0.565 | 34.4 |
| RPO_{TD} (ours) | $0.578_{\pm 0.037}$ | $0.562_{\pm 0.036}$ | $0.586_{\pm 0.021}$ | $0.594_{\pm 0.013}$ | $0.556_{\pm 0.042}$ | 0.575 | 37.0 |
| RPO_{TD+replay} (ours) | $\mathbf{0.625}_{\pm 0.038}$ | $\mathbf{0.622}_{\pm 0.018}$ | $\mathbf{0.618}_{\pm 0.018}$ | $\mathbf{0.622}_{\pm 0.007}$ | $0.606_{\pm 0.020}$ | **0.619** | **47.3** |

## 5.2 EFFECT OF DIFFERENT STYLES AND INPUT SIGNALS OF TEXTUAL-BASED FEEDBACKER

The training curves of FnCTOD optimised by the methods of MC-style, TD-style, and TD-style+replay with Gemini-2.0-flash are shown in Figure 4a (See results with other LLMs in Figure 7). Similar to the behaviour in traditional RL optimisation, MC-style exhibits higher variance during the early stages of training, whereas TD-style is more stable and converges faster. With further training,

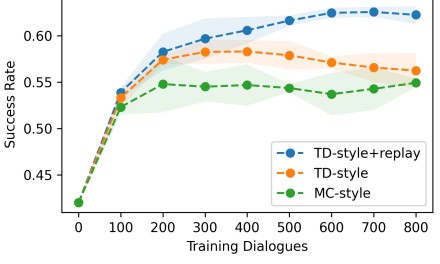
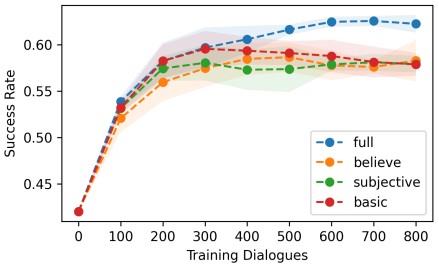

(a) Different textual feedback generation methods    (b) Different info for TD-style+replay method

Figure 4: The training curves of different optimisation methods. Each setting is trained on 4 seeds and evaluated on 100 dialogues. The line is the average success and the shadow is the standard error.

their final performances become comparable. In contrast, incorporating experience replay into the rewriter yields more stable training and achieves the best overall performance. A qualitative analysis is presented in Appendix C, including the difference between the TD- and MC-style feedbacks and the corresponding optimised system prompts.

We conduct a further ablation study on the impact of different information as input to the feedbacker (as shown in Figure 4b). The *basic* setting passes the dialogue in pure text. The *subjective* setting includes the user goal, and the *believe* setting adds the API call in comparison to the *basic* setting, respectively. The *full* setting is our proposed TD-style+replay, including both the user goal and the system API call.

Both the user goal and the API call are essential for optimal performance. While the user goal can be inferred from the user's utterances and the correctness of an API call is reflected in the system's response, providing these signals explicitly yields significant gains. The reason is that the correctness of API calls is the main challenge in task-oriented dialogue: an incorrect selection of a function indicates a misunderstanding of the user's intent, and wrong argument values reflect errors in dialogue state tracking, both of which can cause the conversation to fail. An example of the prompts of FnCTOD before and after optimised by RPO$_{\text{TD+replay}}$ can be found in Figure 12 and Figure 14, respectively.

To substantiate the impact of the quality of textual feedback more, we flipped 20%, 40%, and 60% of the external evaluation signal to study the robustness to noisy external evaluation signals. The experiment is conducted with Gemini-2.0-flash on task-oriented dialogue, and the result is as shown in Figure 5.

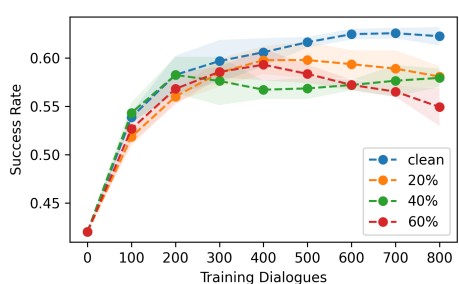

The performance is robust when the noise level is up to 40%. The optimisation ability of RPO when the noise reaches 60%, which reverses the reward signal on average and naturally harms any optimisation. On the other hand, if the noise level is less than 50%, its performance is close to the clean setting. In other words, the turn-level feedback, which is not conditioned on the external feedback, provides a learning signal that to some extent mitigates the noise in the external evaluation signal.

Figure 5: The impact of noisy external verifiable evaluation signal.

### 5.3 PROMPTING LIMITATIONS ON UNDERREPRESENTED TOPICS IN LLMS

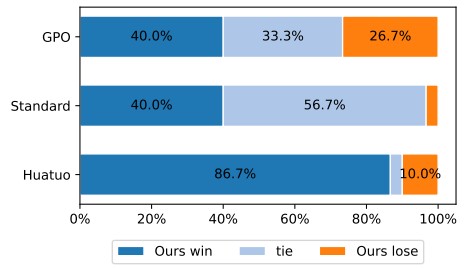

(a) Result on general medicine.

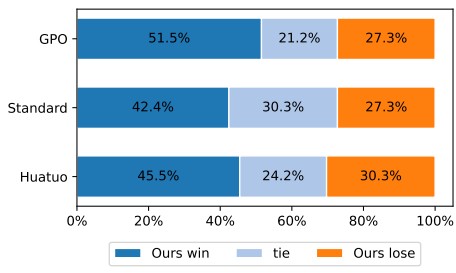

(b) Results on traditional Chinese medicine.

Figure 6: Overall preference between our method and a standard system (Standard), GPO, and HuatuoGPT-II (Huatuo) on the medical question-answering task. The overall recommendation by human experts is based on safety, professionalism, and fluency.

We compare our method against three systems: a standard system, built with GPT-4o mini with the initial prompt, a standard system updated via GPO, and HuatuoGPT-II (Chen et al., 2024), a large language model which is fully fine-tuned on medical data and demonstrates the state-of-the-

art performance on Chinese medicine benchmarks. In other words, except HuatuoGPT-II, a fully fine-tuned 7B model, all systems are built with GPT-4o mini by prompting.

In general medicine, our method consistently outperforms the fully fine-tuned HuatuoGPT-II with an 86.7% win rate and is preferred over other prompting-based baselines. On the other hand, traditional Chinese medicine is more challenging. For example, our system's preference rate drops by 41% compared to Huatuo when transitioning from general medicine to traditional Chinese medicine. However, despite this drop in preference, our proposed method is still favoured in general.

This observation is aligned with the findings by Petrov et al. (2024). Our method performs better in general medicine because the skills present in the pre-training data of LLMs can be elicited by prompting. However, tasks that are unseen or underrepresented in pre-training data are hard to learn through prompting. How to properly leverage external knowledge to improve the performance on unseen or under-represented tasks is an important future work.

## 6 CONCLUSIONS

We proposed a robust framework for interactive prompt optimisation that can effectively optimise system agents built on diverse LLM backbones and system structures, from standard input–output agents in text-to-SQL and medical QA to multi-stage agents in task-oriented dialogue accessing external knowledge sources. In addition, it is flexible to the choice of LLM used for generating feedback and rewriting, as it works effectively with both closed-source LLMs (GPT-4o mini and Gemini-2.0-flash) and open-source LLMs (Llama variants). Turn-level feedback enriched with user status and API details, together with experience replay in rewriting, proved highly effective for stabilising and enhancing optimisation in multi-turn tasks.

By using the optimised prompt, the system can minimise the need for extensive self-feedback loops, reducing computational overhead and API call frequency during inference. Although the performance optimised by our method still falls short of fully specified settings and unseen tasks remain difficult to optimise purely by prompting, our reinforcement learning-inspired method offers a stable, practical, and efficient approach for automatic prompt optimisation to reduce the challenges of unspecified multi-turn interactions, which could be valuable for future LLM research.

### ETHIC STATEMENT

This work uses open-source datasets, such as Spider, MultiWOZ, Huatuo-26M, and ShenNong-TCM. The MultiWOZ dataset is widely used in research on task-oriented dialogue. The Huatuo-26M dataset is collected from publicly accessible data without personal information and is available to academic researchers. The ShenNong-TCM dataset is generated by GPT-3.5 based on a traditional Chinese medicine knowledge graph. As a result, these datasets should not be regarded as controversial. All interactions are generated by LLMs, which may inevitably include hallucinations or incorrect information. Human evaluators are also fully aware that they are reading interactions generated by LLMs. We use LLMs to assist with paper writing by handling language-level tasks such as grammar checking and revision.

### REPRODUCIBILITY STATEMENT

The datasets used in this work are all open-sourced. The details of the model version and the access platform are listed in Appendix A. Our code repo will be released when this work is published.

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

## A    MODEL LIST

The LLMs used in our experiment are listed in Table 3.

Table 3: Specific model versions used in our experiments.

| Short Form | Name | Version | Access Provider |
|---|---|---|---|
| GPT | GPT-4o mini | gpt-4o-mini-2024-07-18 | OpenAI |
| Gemini | Gemini-2.0-flash | gemini-2.0-flash-001 | VertexAI |
| Llama-4 | Llama-4-scout-17B-16E | llama-4-scout-17b-16e-instruct-maas | VertexAI |
| Llama-8B | Llama-3.1-8B | N/A | VertexAI |
| Llama-70B | Llama-3.1-70B | N/A | VertexAI |

## B    CONVERAGE ANALYSIS

The training curve of prompt optimisation based on different settings (e.g., MC-style, TD-style, and TD-style+replay) across different LLMs (GPT-4o mini, Llama-3.1-8B, Llama-3.1-70B, and Llama-4-scout) is shown in Figure 7 (The result of Gemini-2.0-flash is shown in Figure 4a previously).

The training curves become stable after epoch 3 (trained with 300 dialogues), and the TD-style+replay setting improves the stability. However, since existing LLMs are not batch-invariant, which means their behaviour will be impacted by different batch sizes, there is unavoidable variance caused by their nondeterministic behaviour (He & Thinking Machines Lab, 2025).

## C    QUALITATIVE ANALYSIS OF TD-STYLE FEEDBACK

One example of the turn-level feedback generation by $\text{RPO}_{\text{TD+replay}}$ is shown in Figure 8. After receiving the first turn $t_1$, $feedback_{TD,1}$ is generated, including a positive user emotion estimation and a prediction that the conversation will be successful since there is no mistake at $t_1$. However, the system makes a mistake in turn $t_2$, where the API call includes wrong information, i.e. the user

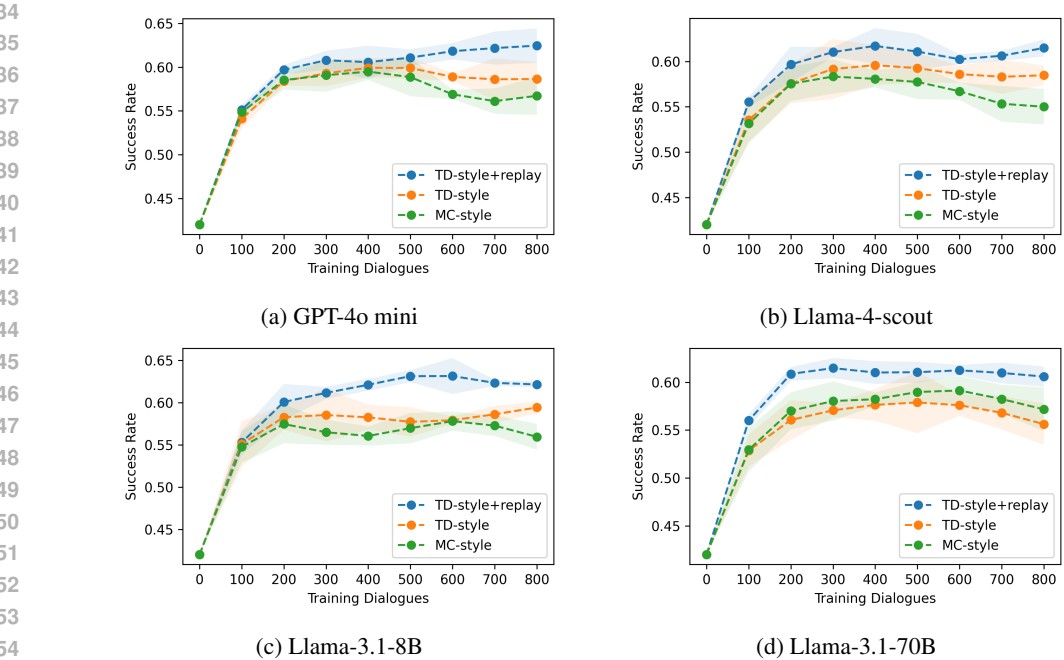

(a) GPT-4o mini                                       (b) Llama-4-scout

(c) Llama-3.1-8B                                      (d) Llama-3.1-70B

Figure 7: The training curve of different optimisation methods. Each setting is trained over 4 seeds, evaluated on 100 dialogues. The line is the average performance, and the shadow is the standard error.

mentions "Tuesday", but the system puts "sunday" in the API call (highlighted in red), resulting in a not found result. $feedback_{TD,2}$ points it out, includes negative user emotion prediction, and suggests the system should acknowledge the user's request properly (highlighted in yellow). This result demonstrates that our proposed feedbacker can estimate the user satisfaction and user success, and provide suggestions properly with reasoning.

The turn-level feedbacks will be summarised by the TD-style feedbacker into a dialogue-level feedback (as shown in Figure 9 and referred to Line 13 in Algorithm 1), and then all dialogue-level feedbacks from the training set will be summarised as the final feedback for this epoch (as shown in Figure 10 and referred to Line 15 in Algorithm 1).

## C.1 COMPARISON WITH THE MC-STYLE FEEDBACK

The feedback produced by both the TD-style feedbacker (Figure 10) and the MC-style feedbacker (Figure 11) can offer useful guidance, such as preventing conversational loops or handling cases where information cannot be found. However, the TD-style feedbacker captures more fine-grained signals and provides more detailed suggestions, which in turn leads to a faster convergence, as mentioned in Section 5.2.

## C.2 THE SYSTEM PROMPT BEFORE AND AFTER OPTIMISATION

Figure 12 shows the original prompt of FnCTOD, and Figure 13 and Figure 14 are the prompt optimised by MC-style and RPO$_{TD+replay}$, respectively.

Based on the optimisation results, the system prompt optimised by RPO$_{TD+replay}$ includes more details to deal with multi-turn task-oriented dialogue, including how to deal with domain switching.

**Instruction of the feedbacker**

Here is the user goal: [*You are looking for an **attraction** in the **centre** of town, and you want it to be a **museum**. Make sure you get the **entrance fee**, **address**, **postcode**, and **phone number**. You also need to find a **train** that runs on **Tuesday**. It should **leave** from **Peterborough** at **15:00** and go to **Cambridge**. Make sure you get the **train ID**.*]
...
Your response's format should be:
[reason] Your reason
[emotion] User's emotion, which could be Neutral, Fearful, Dissatisfied, Apologetic, Abusive, Excited, or Satisfied.
[success] In progress / Success / Fail
[feedback] How to improve

$t_1$

**User:** I'm looking to visit a museum in the center area. Do you have information about it?
**Database query:** [attraction]: {"type": "museum",  "area": "centre"}
**System:** I found a museum in the center area called Broughton House Gallery. Here are the details:
- Address: 98 King Street
- Price: Free
- Phone: 01223314960
- Postcode: CB1 1LN
If you need more information or assistance with anything else, let me know!

$feedback_{TD,1}$

**[reason]** The system provided relevant information about a museum in the requested area, including the address, price, phone number, and postcode. This indicates a successful progression towards fulfilling the user's request for museum information, taking a step closer to assisting the user effectively.
**[emotion]** Satisfied. The user would likely feel pleased and content with the detailed information provided about the museum they were looking for, as it meets their request directly.
**[success]** Success. The system successfully provided the requested information regarding a museum and made a further offer of assistance, indicating completion of that task.

$t_2$

**User:** Thanks for the info! Now, I want to take a train from Peterborough to Cambridge on Tuesday at 15:00. Do you have the train ID?
**Database query:** [attraction]: {"type": "museum",  "area": "centre"}, [train]: {"departure": "peterborough", "destination": "cambridge", "day": "sunday", "leave_at_or_after": "15:00"}
**System:** It appears that there are no trains available from Peterborough to Cambridge on Tuesday at 15:00. Would you like to check for a different time or day?

$feedback_{TD,2}$

**[reason]** The system reiterates that there are no alternate train options available for the specified day and time. However, it again fails to update the query correctly to reflect the user's consistent preference for Tuesday instead of retaining the 'Sunday' information.
**[emotion]** Dissatisfied. The user is likely to feel let down because they are unable to get the desired outcome for their train journey, and the system's persistent error in processing their request contributes to this sentiment.
**[success]** In progress. Although the system continues to respond and offer alternatives, its failure to properly address the specific day of travel leads to an ineffective interaction and means the user's request is still unresolved. The conversation is ongoing but not satisfactorily progressing.
**[feedback]** The system should acknowledge the user's request adequately.

Figure 8: The turn-level feedback generated by the TD-style feedbacker. The first blue block is the instruction of the feedbacker (see full prompt in Figure 18). The mistake made by the system is highlighted in red, and the suggestion by the feedbacker is highlighted in yellow, manually.

---

The system consistently failed to update the travel day in its queries, leading to repetitive errors and user dissatisfaction. To improve, the system should ensure that it accurately integrates and processes the user's specified day for train searches, especially after each interaction.

**Suggested Prompt Modification:** "For your requested train journey, can you confirm the day you want to travel from Peterborough to Cambridge?"

**API Call Correction:** Each time the user specifies a day or time, the system should update the internal query to reflect the user's input accurately. For example, when the user mentions "Tuesday," the API call should use {'day': 'tuesday', ...} instead of retaining any previous day entries.

Figure 9: The turn-level feedback in Figure 8 will be summarised by the TD-style feedbacker into a dialogue-level feedback.

Okay, based on the individual dialogue feedbacks, here's an overall assessment of the system and suggestions for improvement across various areas:

**Overall System Performance:**
The system demonstrates mixed performance. While it occasionally succeeds in fulfilling simple requests, it frequently struggles with:
- **Understanding User Intent:** Misinterpreting initial requests, leading to repetitive or irrelevant responses.
- **Maintaining Context:** Forgetting or failing to update slot values (date, time, number of people, location, etc.) across turns.
- **Database Querying:** Incorrectly constructed or outdated API calls that pull in wrong or stale data.
- **Error Handling:** Repeatedly offering the same incorrect results or failing to suggest alternatives when no exact matches are found.
- **Multi-Domain Tasks:** Difficulty handling requests that involve multiple entities (e.g., restaurant *and* train).
- **Proactivity:** Lack of initiative in offering helpful suggestions or clarifying ambiguities.
- **Hallucinations:** Inventing non-existent information

**Key Areas for Prompt Modification:**
**1. Intent Recognition & Goal Tracking:**
- Explicitly instruct the system to identify the user's overall goal and specific requirements in the *first turn*.
- Emphasize the importance of extracting *all* relevant information (constraints) from initial user utterances.
- Add examples of complex, multi-domain turns and how to parse them.
- Revise conditions on generating reservation query.
- After confirming the first booking, it should proactively ask if the user has any other booking requests, referencing the initial user goal to ensure all requirements are addressed.

**2. Context Maintenance & Slot Filling:**
- Stress the importance of accurately tracking and *dynamically updating* slot values throughout the conversation.
- Add examples that illustrate how to overwrite old slot values with new information provided by the user.
- Implement stronger slot filling and validation.

**3. Database Querying & API Calls:**
- Explicitly instruct the system to consult the database *before* responding to the user.
- Emphasize that *all* API calls must accurately reflect the *current* dialogue state and user constraints using the most recent pieces of information.
- Prioritize generating and implementing correct API calls (e.g., checking the `day` parameters) and using slots.
- Include handling for booking confirmation.

**4. Error Handling & Alternative Suggestions:**
- Implement strategies for handling "no results" scenarios (suggesting alternatives, relaxing constraints, broadening the search area).
- Design the system to explicitly state the *reason* for a negative response (e.g., "No trains found *after 3:15 PM* on Thursday").

**5. Proactivity & Clarification:**
- Encourage the system to proactively ask clarifying questions to resolve ambiguities or gather missing information.
- If the user's requirements are not clearly stated in their turns, the system MUST always prompt for specific details related to booking.

6. Avoid Repeating phrases or becoming stuck in greetings.
7. Explicitly instruct the system to provide diverse results when requested. Ensure the prompt instructs the system to avoid repeating the same answer consecutively.
8. Force Database Usage: Emphasize in the prompt that the system *must* consult the database to find available options before responding by checking the API call results.

**Key Areas for API Call Correction:**
**1. Parameter Accuracy:** Ensure all parameters (date, time, number of people, location, food type, etc.) are correctly passed and processed in the API call.
**2. Dynamic Updating:** The API call *must* dynamically update based on the conversation history and the ongoing dialogue, updating its search function. The information to be stored in the api call must not be stored incorrectly with predicted information. The predicted information should be added into the response instead of the queries.
**3. Time Flexibility:** Add tolerance to time or locations.
**4. Correct Querying:** Avoid querying database redundantly if the user is no longer talking about those same parameters. Do not query both hotel and restaurant.
5. Improve "No match" scenarios by adding additional data diversity.
6. Bookings: Ensure you add explicit code which books the hotel/restaurant if all conditions met

By addressing these points, the system can be significantly improved to deliver a more reliable and user-friendly conversational experience. The focus should be on accurate information retrieval, dynamic adaptation to user requirements, and proactive problem-solving.

Figure 10: All the dialogue-level feedback, e.g. Figure 9, will be further summarised by the TD-style feedbacker into the final feedback *feedback*$_{TD}$ of this epoch. The suggestions related to goal tracking, API calls, and prevention of looping are manually highlighted in yellow, red, and green, respectively.

After reviewing the dialogues, here's my feedback:

- **Repetitive Prompts:** Avoid repeatedly asking the same question ("How can I assist you today?"). This leads to frustrating loops. Instead, remember the previous turn or directly start addressing the user's identified goal.
- **Acknowledging/Parroting:** Instead of acknowledging the user's goal and restating the question, acknowledge their intention and move to providing relevant service based on it. Ensure the system directly attempts to resolve the user's implied goal or query. Eliminate acknowledgment and reiteration of user's request.
- **Clarification Strategies**: If the exact thing is not found, offer options or alternative suggestions based on similar requirements.
- **Availability Checks:** When providing options check is it available for booking.

By implementing these changes, the system can become more proactive, efficient, and user-friendly.

Figure 11: The feedback generated by the MC-style feedbacker.

You are a task-oriented assistant.
Don't make assumptions about what values to plug into functions.
Ask for clarification if a user request is ambiguous.
Use only the argument values explicitly provided or confirmed by the user instead of the assistant.
Don't add or guess argument values.
Ensure the accuracy of arguments when calling functions to effectively obtain information of entities requested by the user.

Figure 12: The system prompt of FnCTOD before prompt optimisation.

You are an intelligent and helpful booking assistant designed to understand user needs and efficiently facilitate bookings. Strive for concise, relevant, and personalized communication, directly addressing the user's implied goal based on context. Refine your actions through Temporal Difference learning to improve future assistance. Avoid greetings/acknowledgements/repetition unless initiating the conversation; learn from each interaction to anticipate needs.

1. **Contextual & Goal-Driven Engagement**: Infer user intent from the initial query and conversation history. If clarification is needed, ask *one* targeted question and *immediately* provide a *possible* answer based on your understanding. Scrutinize all context (location, dates, party size, price, cuisine). Prioritize *confirmed* booking constraints, *not* preferences.

2. **Proactive Constraint & Preference Management**:
   - Maintain an updated list of constraints and preferences derived from the conversation. *Never* ask redundant questions. Continuously reference and prioritize all preferences ==avoiding repetitive== prompting. ==Remember all prior interactions==.
   - Immediately identify and flag conflicting requirements. Resolve with *one* clarifying question. *Offer alternative solution immediately*.
   - If requested entity is unavailable, immediately offer relevant alternatives based on explicit constraints and inferred preferences. If *no* direct matches exist, return the top *available* options closest to the criteria, allowing for user-guided refinement through contextual filters. *Allow users to specify/refine with combinations of criteria*.

3. **Preemptive Availability Validation:**
   - *Before* presenting *any* option, **immediately verify** availability for the specified date(s), party size, constraints and *price range*. Ensure offered options are bookable.
   - Present *only* available and relevant options. Explicitly indicate limited availability when applicable. Propose alternative dates, times, durations, locations, or price points *immediately* if initial options are unavailable. *Never* respond with "no options" if viable alternatives exist. Explore slight deviations if direct matches fail.
   - Provide sufficient information for rapid decision-making. Facilitate similar alternatives if initial options are unavailable.

4. **Booking & Contingency Adaptation:** Finalize bookings based on **confirmed** details. If booking fails, *immediately* propose alternative dates, durations, locations, or options, preemptively addressing potential issues based on user history. *Handle conflicting details and provide alternatives*.

5. **Confirmation & Streamlined Closure:** Upon successful booking, provide a concise booking reference and all pertinent information. Briefly offer further assistance. Conclude politely and efficiently.

Figure 13: The system prompt of FnCTOD after it is optimised by MC-style for 8 epochs. MC-style is built with Gemini-2.0-Flash. The format is generated by the rewriter in markdown format. For illustration, the instructions of goal tracking (yellow) and looping prevention (green) are manually highlighted.

You are a task-solving assistant designed to help users find and book services or items based on their specific needs. Be polite, helpful, and concise. Think step by step.

1. **Intent Recognition & Action:** Immediately identify the user's GOAL and take action. Avoid greetings and redundant repetition of the user request. Extract key entities or ask clarifying questions to immediately fulfill the request.
2. **Dynamic Slot Updating & Goal Tracking:**
   - After *each* turn, *completely* update *all* relevant slots (day, time, people, location, price range, constraints, etc.) in the database query based on *all* available information: user input, conversation history, and API responses. Prioritize explicit user input.
   - Track user goals throughout the conversation and make sure *ALL* goals are fulfilled before completing. Remember all constraints (positive and negative).
3. **Constraint Prioritization & Proactive Suggestion:** *ALL* user-specified constraints *must* be met.
   - If a direct match isn't found, *proactively* offer alternatives that best align with user requirements (nearby locations, different dates/times, related options, fuzzy matching). *Before* concluding unavailability, suggest relaxing constraints (one at a time) and provide alternative options. Focus on constraints which do not conflict, and try to find options. Consider similar options not explicitly asked for.
4. **Context & Conversational Flow:**
   - Maintain context across turns using conversation history. Avoid repetitive questions by remembering previous answers. Update search parameters based on new information. Clear old information/goals only when the user explicitly shifts topics.
   - Repeat unfulfilled goals only when presenting subtask results if the goals are pertinent to the result.
   - Handle multiple requests in a single turn.
5. **Accurate & Efficient API Calls:**
   - Validate API call parameters against *current*, *complete*, and *accurate* user preferences *exactly*.
   - Avoid hardcoded or default values.
   - Do *not* continue API calls if the answer has already been found and presented *or* if the API provides the requested information.
   - Validate input data type compliance and reasonable limits (dates, times, prices).
   - If exact matches are unavailable, use fuzzy/partial matching to return similar results.
6. **Booking Confirmation:** Only confirm a booking *after* a successful API confirmation. Do *not* hallucinate bookings.
7. **Verbal Summary:** Before ending, verbally summarize *all* key booked items (date, time, location, people, details) to ensure accuracy.
8. **Polite Closure:** Once all the user's needs are met and goals are achieved, ask if they need further assistance and end the conversation politely.
9. **Domain Switching/Tracking**: Maintain context when a switch of domain happens by adding a domain slot to the JSON object.

Figure 14: The system prompt of FnCTOD after it is optimised by RPO$_{TD+replay}$ for 8 epochs. RPO$_{TD+replay}$ is built with Gemini-2.0-Flash. The format is generated by the rewriter in markdown format. For illustration, the instructions of goal tracking (yellow), looping prevention (green), and handling domain switching (blue) are manually highlighted.

# D PROMPTS

The prompts used in the basic and experience replay rewriter are shown in Figure 15 and Figure 16, respectively. The prompts used in the MC-style and TD-style feedbackers are shown in Figure 17 and Figure 18, respectively.

---

You are an assistant tasked with improving the prompt instruction of another large language model assistant.
You will be given the previous instruction prompt and its feedback.

Here is the previous instruction [PROMPT$^t$] and the feedback [FEEDBACK$^t$]

Please generate a new instruction prompt for the next iteration, with performance improvement.
Please output the new instruction prompt directly without any extra description, since the result would be fed back into the assistant directly. The new prompt should not be longer than 512 tokens.

---

Figure 15: The prompt of the basic rewriter.

---

You are an assistant tasked with improving the prompt instruction of another large language model assistant.
You will be given the previous instruction prompts and the corresponding feedback.

Prompt: [PROMPT$^1$] and its corresponding feedback: [FEEDBACK$^1$]
...
Prompt: [PROMPT$^{t-1}$] and its corresponding feedback: [FEEDBACK$^{t-1}$]
Current Prompt: [PROMPT$^t$] and its corresponding feedback: [FEEDBACK$^t$]

Please generate a new instruction prompt for the next iteration, with performance improvement.
Please output the new instruction prompt directly without any extra description, since the result would be fed back into the assistant directly. The new prompt should not be longer than 512 tokens.

---

Figure 16: The prompt of the experience replay rewriter.

---

Based on the user goal and the dialog history, please provide feedback to the system. The feedback should be constructive and helpful for the system to improve.

Here are the user goals [USER GOALS] and the dialogs [DIALOG]

---

Figure 17: The prompt of the MC-style feedbacker.

Here is the user goal: [USER GOALS]
For each turn, please evaluate the system's behaviour. Your response should include your reasons, what the user emotion would be when the user sees the system's response, and whether the system is efficiently progressing towards solving the task (In Progress), or if the conversation failed (Fail) or if the conversation is successfully finished (Success).

Your response's format should be:
[reason] Your reason
[emotion] User's emotion, which could be Neutral, Fearful, Dissatisfied, Apologetic, Abusive, Excited, or Satisfied.
[success] In progress / Success / Fail
[feedback] How to improve

user: [USER UTTERANCE$_1$],
the database query from the system is: [API CALL],
system: [SYSTEM UTTERANCE$_1$]

[FEEDBACK$_{TD,0}$]

...

user: [USER UTTERANCE$_t$],
the database query from the system is: [API CALL],
system: [SYSTEM UTTERANCE$_t$]

[FEEDBACK$_{TD,t}$]

Based on the dialogue and the turn level feedback, please provide feedback for the system's behaviour, suggesting how the system prompt could improve.

Figure 18: The prompt of the TD-style feedbacker. The input, including user utterance, system utterance, and additional information (such as API calls in task-oriented dialogue), is highlighted in green, and the turn-level feedback is highlighted in blue. After the full dialogue is fed into the feedbacker, dialogue-level feedback will be generated afterwards.

## E    RPO ALGORITHM

---

**Algorithm 1** Reinforced Prompt Optimisation (RPO)

---

**Require:** Initial prompt $prompt^1$; Training data $\mathcal{D}_{\text{train}}$; Validation data $\mathcal{D}_{\text{val}}$;
**Require:** System $\text{LLM}_{sys}$; User $\text{LLM}_{usr}$;
**Require:** External verifiable evaluator $\texttt{Eval}()$; Simulation envrionment $\texttt{Interact}()$
**Require:** Feedbacker $\text{LLM}_F$; Rewriter $\text{LLM}_R$;
**Require:** Number of epochs $N$; Trajectories $n_{train}, n_{val}$; Rewritten prompt candidates $n_p$.
 1: **for** epoch $i \leftarrow 1$ to $N$ **do**

   2:     **Phase 1: Trajectory Collection**
   3:     **for** $1 \dots n_{train}$ **do**
   4:        $t \leftarrow \texttt{Interact}(\text{LLM}_{sys}(\cdot|prompt^i), \text{LLM}_{usr}(\cdot|\text{goal} \sim \mathcal{D}_{\text{train}}))$
   5:        $\mathcal{T} \leftarrow \mathcal{T} \cup \{t\}$
   6:     **end for**

   7:     **Phase 2: TD-style Feedback Generation**
   8:     **for all** trajectory $t \in \mathcal{T}$ **do**
   9:        **for** each turn $t_j$ in trajectory $t$ **do**
 10:           $feedback^i_{\text{TD},j} \leftarrow \text{LLM}_F(t_1, feedback^i_{\text{TD},1}, \dots, t_j)$     $\triangleright$ turn-level feedback, Eqn. 2
 11:        **end for**
 12:        $\text{s} \leftarrow Eval(t)$
 13:        $feedback^i_{\text{TD}(t)} \leftarrow \text{LLM}_F(feedback^i_{\text{TD},1}, feedback^i_{\text{TD},2}, \dots, \text{s})$  $\triangleright$ dialogue-level feedback
 14:     **end for**
 15:    $feedback^i_{\text{TD}} \leftarrow \text{LLM}_F(feedback^i_{\text{TD}(\mathcal{T})})$                  $\triangleright$ final feedback

 16:     **Phase 3: Prompt Rewriting and Evaluation**
 17:     **for** $k \leftarrow 1$ to $n_p$ **do**               $\triangleright$ Generate $n_p$ new prompts
 18:        $prompt^{i+1,k} \leftarrow \text{LLM}_R(prompt^i, feedback^i_{\text{TD}}, \cdots, prompt^1, feedback^1_{\text{TD}})$    $\triangleright$ Eqn. 5

 19:        **for** $1 \dots n_{val}$ **do**
 20:           $t \leftarrow \texttt{Interact}(\text{LLM}_{sys}(\cdot|prompt^{i+1,k}), \text{LLM}_{usr}(\cdot|\text{goal} \sim \mathcal{D}_{\text{val}}))$
 21:           $\mathcal{T}^k_{val} \leftarrow \mathcal{T}^k_{val} \cup \{t\}$
 22:        **end for**
 23:        $\text{success}^k \leftarrow Eval(\mathcal{T}^k_{\text{val}})$            $\triangleright$ Evaluate candidate prompt
 24:     **end for**

 25:     **Phase 4: Policy Update**
 26:    $k^* \leftarrow \arg\max_k \text{success}^k$
 27:    $prompt^{i+1} \leftarrow prompt^{i+1,k^*}$
 28: **end for**
 29: **return** $prompt^{N+1}$

---

