# OpenReview forum: "Prompt reinforcing for long-term planning of large language models"
_ICLR.cc/2026/Conference — Submitted to ICLR 2026_

### Official Review · Reviewer_mY4n · 2025-10-30

**Soundness:** 3
**Presentation:** 3
**Contribution:** 3
**Rating:** 6
**Confidence:** 3

**Summary:**

This paper generalizes the idea of self-refinement to the instruction prompt and conditions on step-level feedback from the model at intermediate steps. They demonstrate that this prompt learning approach significantly outperforms other single-turn approaches for optimizing prompts in accordance with feedback.

**Strengths:**

1. The paper demonstrate sthat this approach is more cost-effective and performant than existing prompt optimization approaches
2. The idea proposed in the paper is simple, elegant, and well formalized, such that any researcher could implement this on their own tasks
3. The paper clearly explains with mathematics why this approach for fine-grained feedback might be more appropriate than single-turn feedback
4. Using this method is more cost-effective for optimization than training-based approaches

**Weaknesses:**

1. Prompt optimization is useful for improving performance but struggles when LLMs or humans can give useful feedback, but what should practictioners do if the feedback is not useful
2. The paper shows improvement, but this reviewer doesn't find any qualitative analysis of the feedback generated
3. It is unclear how this might scale as the trajectory gets longer. If feedback is applied to every single step this could become very expensive in terms of API costs.

**Questions:**

1. How do we ensure that the feedback given to the models is properly aligned and accurate? The fact that the reward improves is perhaps enough evidence, but do you have qualitative examples?
2. How many trajectories are taken into consideration when doing this? Are the prompts optimized within trajectories or across trajectories? E.g., do we take feedback from multiple trajectories on the same prompt and use this or is it only within the same trajectory that we optimize?
3. Is the feedback incorporated into the instruction prompt in-flight or only at the end of the trajectory? For example, if the model makes a mistake on step x and gets feedback, is the instruction immediately changed to address this or is it only at the end of the turn?
4. From this reviewer’s perspective, prompt-based optimization has fundamental limitations in regards to scaling or teaching models wholly new ideas. Do the authors have any ideas on how this type of approach could be combined with parameter updates to do this sort of optimization, or be used in a domain where an LLM or human might not be suitable to give good feedback?

---

> ### Author Response · Authors · 2025-11-21
>
> # The impact of feedback quality
> > Weakness 1: Prompt optimization is useful for improving performance, but struggles when LLMs or humans can give useful feedback. What should practitioners do if the feedback is not useful?
>
> This is an excellent point, thank you for raising this. In Section 5.2 (Figure 4b), we already demonstrated RPO's robustness to incomplete information in the feedbacker input. By removing specific information, e.g. removing the user goal in the *believe* setting, removing the API call in the *subjective* setting, or removing both the user goal and API call in the *basic* setting, RPO can still improve the system with these suboptimal feedback signals, and the cleaner and more informative feedback signal (the full setting) can reach better performance naturally.
>
> To substantiate this more, we have conducted a **new** experiment on robustness to noisy external evaluation signal, where we flipped 20%, 40%, and 60% of the external evaluation signal (please see the last part of Section 5.2 in our revision). The experiment is conducted with Gemini-2.0-flash on task-oriented dialogue, and we are very excited to report the results:
>
> | Noise level | Success |
> | :--- | :---: |
> | **clean** | $0.622 \pm 0.018$ |
> | **20%** | $0.581 \pm 0.021$ |
> | **40%** | $0.579 \pm 0.018$ |
> | **60%** | $0.549 \pm 0.038$ |
>
> The performance is robust when the noise level is up to 40%. The optimisation ability of RPO declines when the noise reaches $60\%$, which reverses the reward signal on average and naturally harms any optimisation. On the other hand, if the noise level is less than 50%, its performance is close to the clean setting. In other words, the turn-level feedback, which is not conditioned on the external feedback, provides a learning signal that, to some extent, mitigates the noise in the external evaluation signal.
>
> In summary, our proposed RPO is robust to noisy or incomplete reward signals, but providing more informative and clean info to the feedbacker can generate better textual feedback, which optimises the system agent further.
>
> > Question 4c: How to use in a domain where an LLM or human might not be suitable to give good feedback?
>
> In addition to the subjective feedback signals, e.g. human/LLM feedback, our method also includes objective, verifiable reward signals, e.g. task success for task-oriented dialogue and exact match for text-to-SQL, as shown in Figure 2 and Figure 3. In this way, we can make the LLM agent focus not only on the feedback from humans or LLM (which may be informative but not perfectly aligned or noisy), but also on the objective evaluation signals.

---

> ### Author Response · Authors · 2025-11-21
>
> # Qualitative analysis: Comparison of the feedback and the optimised prompts from different prompt optimisation methods
> > Weakness 2: The paper shows improvement, but this reviewer doesn't find any qualitative analysis of the feedback generated
>
> > Question 1: How do we ensure that the feedback given to the models is properly aligned and accurate? The fact that the reward improves is perhaps enough evidence, but do you have qualitative examples?
>
> We add a qualitative analysis for the turn-level feedback in Appendix E and for updated system prompts generated by different feedback signals in Appendix C.
> This analysis demonstrates that the LLM-based feedbacker can estimate the user satisfaction and user success, and provide suggestions properly with reasoning. Please check our revision for further details.
>
> In addition, as shown in Figure 2, Line 13 in Algorithm 1, the dialogue-level textual feedback is also conditioned on an external evaluator, e.g. *task success* in task-oriented dialogue and *exact match* in Text-to-SQL. In other words, by leveraging both text-based feedback and an objective, verifiable evaluator, we balance the final feedback to be informative and aligned with objective evaluation results.
>
> # Scalability
> > Weakness 3: It is unclear how this might scale as the trajectory gets longer. If feedback is applied to every single step, this could become very expensive in terms of API costs.
>
> In our experiment on task-oriented dialogue, the feedbacker’s input, which includes the user utterance, system API call, and system response, averages $128.9 \pm 2.6$ tokens per turn, while the turn-level feedback averages $117.5 \pm 1.8$ tokens. Modern LLMs such as Gemini-2.0-flash, with a 32k-token context window, can therefore handle more than 130 turns. However, processing such long sequences may still benefit from intermediate summarisation to mitigate the lost-in-the-middle issue. In principle, one *does not* have to generate feedback at every turn; it can be done at every $k$ turns if the cost of API calls is an issue, but the method is general enough to allow for this modification. Further modifications are also possible, inspired by eligibility traces in TD-learning when one wants to utilise each data point as much as possible. The investigation of these we leave for future work.

---

> ### Author Response · Authors · 2025-11-21
>
> # Implementational details
> > Question 2: How many trajectories are taken into consideration when doing this? Are the prompts optimized within trajectories or across trajectories? E.g., do we take feedback from multiple trajectories on the same prompt and use this or is it only within the same trajectory that we optimize?
>
> As mentioned in Section 4.3, the feedbacker takes $10$ trajectories into consideration during generating the feedback. In other words, we take feedback from $10$ trajectories (interacting with different user goals) with the same system prompt, then these dialogue-level feedbacks will be summarised into a final feedback as shown in Figure 2.
>
> > Question 3: Is the feedback incorporated into the instruction prompt in-flight or only at the end of the trajectory? For example, if the model makes a mistake on step x and gets feedback, is the instruction immediately changed to address this or is it only at the end of the turn?
>
> As shown in Equation 2 and Figure 1, the feedback is generated *after* the interaction is collected. As a result, the system prompt remains the same while interacting with the environment.
>
> We added the algorithm in our revised version to clarify the implementation of RPO.
>
> # The limitation of prompt optimisation (in-context learning)
> > Question 4a: From this reviewer’s perspective, prompt-based optimization has fundamental limitations in regards to scaling or teaching models wholly new ideas.
>
> As mentioned in the last part of Section 2, the limitation of prompt-based optimisation is that transformers struggle to learn new tasks solely through prompting [1]. This observation also aligns with our experiments, i.e. although the feedback is generated by human experts (doctors in general medicine and traditional Chinese medicine), the improvement on traditional Chinese medicine by our prompt optimisation is still limited (see Section 5.3).
>
>
> > Question 4b: Do the authors have any ideas on how this type of approach could be combined with parameter updates to do this sort of optimization?
>
> There are several works which include textual feedback into a refinement model with parameter updates:
> * Jin et al. (2023) investigate the influence of incorporating different types of information, such as the original prompt, critique, and revised response, during training a refinement model [2].
> * Chet et al. (2024) train a refinement model based on a dataset, which includes incorrect samples with error descriptions, improvement suggestions, and expected outputs, to generate improved responses conditioned on the erroneous outputs and corresponding feedback [3].
> * Lloret et al. (2024) further demonstrate that fine-tuning a model conditioned on textual feedback enables controllable behaviour modification through varying feedback inputs [4].
>
> Incorporating our fine-grained feedback signal into these works would be an interesting future work.
>
> [1] When Do Prompting and Prefix-Tuning Work? A Theory of Capabilities and Limitations (Petrov et al., ICLR 2024)
>
> [2] Data-Efficient Alignment of Large Language Models with Human Feedback Through Natural Language (Jin et al., NeurIPS 2023)
>
> [3] Learning from Natural Language Feedback (Chen et al., TMLR 2024)
>
> [4] Towards Aligning Language Models with Textual Feedback (Lloret et al., EMNLP 2024)

---

> > ### Comment · Reviewer_mY4n · 2025-11-26
> > **Thank you**
> >
> > I would like to thank the authors for addressing my concerns and for the new experiment provided, I have updated my score to reflect this!

---

### Official Review · Reviewer_qCy2 · 2025-10-31

**Soundness:** 2
**Presentation:** 3
**Contribution:** 3
**Rating:** 6
**Confidence:** 3

**Summary:**

This paper introduces Reinforced Prompt Optimisation (RPO), a  reinforcement learning–inspired framework for improving long-term planning and consistency of large language models (LLMs) in multi-turn interactions.

**Strengths:**

* Treating prompts as *optimizable parameters* under a reinforcement learning paradigm is both interesting and practical, especially for API-constrained models. Unlike prior prompt optimisation work, RPO explicitly models *temporal feedback* across multiple dialogue turns, addressing a critical weakness of current LLMs.
* The experiments span diverse tasks (SQL, dialogue, medical QA) and model families, demonstrating robustness and generalization.

**Weaknesses:**

* One limitation is the limited theoretical analysis. While the method draws inspiration from RL, there is no formal definition and analysis on how to relate the concepts of RPO to state, action and value functions.
* Most results rely on task success or human preference; it might be interesting to study more on the roles of different LLM players (rewriter and feedbacker). For example, what is the quality of the feedbacker? Are rewards and values generated in reasonable ways?

**Questions:**

1. Equation 3, is the value function actually optimal value function $V^*$?
2. Figure 4, the text is too small to read;
3. Figure 10, is the turn-level feedback still generated given the full dialogue? Why not generate feedback per turn?
4. How sensitive is RPO to the feedback quality or the LLM backbone used as the feedbacker? Would weaker models still yield improvement?
5. How do you prevent prompt degradation over iterations? Is there a stopping criterion or performance monitor to avoid over-editing?
6. What are the failure modes*observed during optimisation? Are there cases where RPO overfits to user trajectories or amplifies LLM hallucinations?
7. Could future work explore numerical reward modeling (e.g., via LLM-generated scores) to combine textual feedback with scalar signals?

---

> ### Author Response · Authors · 2025-11-21
>
> We are pleased to see that you acknowledge our contribution to incorporating temporal feedback to enhance the multi-turn capabilities of LLM-based agents and the comprehensiveness of our experimental setting, which demonstrates the robustness and generalisability of our proposed method.
> In addition, thank you for raising the readability issue in Question 2, and we have increased the font size of Figure 4 in the revision.
>
> # Theoretical analysis: linking RPO to reinforcement learning concepts
> > Weakness 1: One limitation is the limited theoretical analysis. While the method draws inspiration from RL, there is no formal definition and analysis on how to relate the concepts of RPO to state, action and value functions.
>
> As introduced in Section 3.1 (L221-224) and equation 3, we address the prompt optimisation with a reinforcement learning algorithm, where the policy parameter is the system instruction, the state is the encoding of the dialogue so far, the system's action is the response of the LLM, and the value function is approximated by the feedbacker. Taking the reward signal from the feedbacker and external evaluator, the rewriter applied the textual gradient to the system prompt for optimisation.
>
> We will clarify the link between the reinforcement learning algorithm and RPO in our revision.
>
> > Question 1: Equation 3, is the value function actually optimal value function V*?
>
> Our proposed RPO is analogous to policy-based reinforcement learning, i.e. we do not learn a value network but optimise the system prompt. As a result, the value function is not the optimal value function.
>
> # Implementational details
> > Question 3: Figure 10, is the turn-level feedback still generated given the full dialogue? Why not generate feedback per turn?
>
> As shown in Equation 2, the turn-level reward is generated interactively, i.e., the feedback agent is first instructed to generate feedback, then receives each turn from the trajectory and generates the response. In other words, the turn-level feedback is generated per turn, and the feedbacker does receive the full trajectory when generating feedback for the final turn naturally. One example from our experiment log is shown in Figure 14.

---

> ### Author Response · Authors · 2025-11-21
>
> # Ablation study on Rewriter and Feedbacker
> > Weakness 2: it might be interesting to study more on the roles of different LLM players (rewriter and feedbacker). For example, what is the quality of the feedbacker? Are rewards and values generated in reasonable ways?
>
> To study the impact of meta-prompting agents, we have already conducted several ablation studies, including:
> * the choice of base model for meta-prompting agents:
>   > Question 4: How sensitive is RPO to the feedback quality or the LLM backbone used as the feedbacker? Would weaker models still yield improvement?
>
>   Based on our experiment in the second part of Section 5.1 (Table 2), there is *no statistically significant difference* in performance for our proposed method $RPO_{TD+replay}$ *between the different choices of the LLMs* for the meta-prompting agent. This demonstrates the robustness and generalisability of our proposed method.
> * the prompt strategy of rewriter (Section 5.1):
>
>   The experience replay rewriter can reach a *better performance* as shown in Table 1 and Table 2, and can provide a *more stable training curve* as shown in Figure 4a in Section 5.2 and Figure 6 in Appendix B.
> * incomplete input information for feedbacker:
>
>   In Section 5.2, we provide **incomplete information** to the feedbacker, e.g. removing the user goal in the *believe* setting, removing the API call in the *subjective* setting, or removing both the user goal and API call in the *basic* setting. As shown in Figure 4b, RPO can still improve the system, which demonstrates the *robustness to lower quality of feedback* signal, and the most informative input (full setting) can reach the optimal performance.
>
> * *NEW* noisy signal from external evaluator:
>
>   We have conducted a *new* experiment on robustness to **noisy external evaluation signal**, where we flipped 20%, 40%, and 60% of the external evaluation signal (please see Figure 5 in Section 5.2 in our revision). The experiment is conducted with Gemini-2.0-flash on task-oriented dialogue, and the result is as follows.
>
>   | Noise level | Success |
>   | :--- | :---: |
>   | **clean** | $0.622 \pm 0.018$ |
>   | **20%** | $0.581 \pm 0.021$ |
>   | **40%** | $0.579 \pm 0.018$ |
>   | **60%** | $0.549 \pm 0.038$ |
>
>   The performance is robust when the noise level is up to 40%. The optimisation ability of RPO declines when the noise reaches 60%, which reverses the reward signal on average and naturally harms any optimisation. On the other hand, if the noise level is less than 50%, its performance is close to the clean setting. In other words, the turn-level feedback, which is not conditioned on the external feedback, provides a learning signal that, to some extent, mitigates the noise in the external evaluation signal.
>
>
> > Question 5: How do you prevent prompt degradation over iterations? Is there a stopping criterion or performance monitor to avoid over-editing?
>
> Thank you for raising this question. In our experiment setting, the rewriter will generate multiple prompt candidates, and these candidates will be evaluated over the validation set (Section 4.3). The max epoch number is set manually.
> Although the experience replay rewriter can make the optimisation more stable, it is still a future work direction to provide a more controllable and stable prompt optimisation setting.
>
> > Question 6: What are the failure modes*observed during optimisation? Are there cases where RPO overfits to user trajectories or amplifies LLM hallucinations?
>
> One common failure mode is that the improper prompt style of the feedbacker introduces non-ideal feedback signals. For example, the Automatic prompt optimisation (APO) [1] is designed for classification tasks (see their implementation of `_get_gradients` in their repo [2]), which may generate feedback mistakenly suggesting a dialogue system to be a classifier.
> However, RPO includes textual feedback and scaler signal from an external evaluator, which can make the new system prompt still grounded with verifiable evaluation signals. In addition, the feedbacker will check the parameters in the API call, which further makes sure the system does not hallucinate non-existing values (as shown in Figure 14).
>
> [1] Automatic Prompt Optimization with "Gradient Descent" and Beam Search (Pryzant et al., EMNLP 2023)
>
> [2] https://github.com/microsoft/LMOps/blob/main/prompt_optimization/optimizers.py#L58

---

> ### Author Response · Authors · 2025-11-21
>
> # Future directions
> > Question 7: Could future work explore numerical reward modelling (e.g., via LLM-generated scores) to combine textual feedback with scalar signals?
>
> That is indeed a very interesting topic to investigate how to incorporate LLM-generated numerical scores alongside textual feedback. Our intuition is that a key challenge lies in the fact that scalar outputs from LLMs are often poorly calibrated. However, recent work shows that reasoning-based prompting can improve the consistency and reliability of LLM judgments. Therefore, combining our informative textual feedback with such calibrated scoring mechanisms could potentially yield a more effective numerical reward signal.

---

> > ### Comment · Reviewer_qCy2 · 2025-11-26
> >
> > Thank all authors for answering my questions!

---

### Official Review · Reviewer_n39h · 2025-10-31

**Soundness:** 3
**Presentation:** 3
**Contribution:** 2
**Rating:** 2
**Confidence:** 4

**Summary:**

The authors propose a prompt optimization technique designed to mitigate performance degradation of large language models (LLMs) in multi-turn dialogue. The proposed method improves textual prompts without using gradients, making it cost-efficient and applicable to API-only models. The experiments demonstrate strong performance across multiple aspects.

**Strengths:**

The proposed method is easy to follow and can be applied quickly.

The experiments are solid and well support the authors’ claims.

The writing quality is good.

**Weaknesses:**

Lack of comparison with RL-based prompt generation methods.
The paper compares only against text-gradient-based approaches, but there also exist fully RL-based methods such as StablePrompt [1], which trains a small rewriter model using reinforcement learning. Such methods are generally reported to outperform text-gradient approaches and offer more stable training. Moreover, StablePrompt can be applied to API-based models since it does not directly optimize the system agent. The paper should at least demonstrate that the proposed method outperforms RL-based methods like StablePrompt in some scenarios.

Lack of quantitative comparison on API call costs.
Because RPO uses a replay buffer to reuse prompt–feedback pairs over multiple epochs, it could lead to increased API calls and token usage. While the authors claim efficiency, they do not provide quantitative evidence showing how many more API calls or tokens RPO consumes compared to APO or GPO.

Insufficient statistical reliability of results.
Figure 4 shows high variance in RPO’s performance. Moreover, text-gradient-based methods are known for their instability and high variance during training. Reporting standard deviations in Tables 1 and 2 would help alleviate concerns regarding reliability.

Missing ablation study for the Feedbacker and Rewriter components.
Although the paper highlights the importance of the Feedbacker and Rewriter, no experiments are conducted to replace or analyze them individually. The paper provides insufficient explanation of the prompts used for generating text gradients.

Reference
[1] Kwon et al. "StablePrompt: Automatic Prompt Tuning using Reinforcement Learning for Large Language Models." Proceedings of the 2024 Conference on Empirical Methods in Natural Language Processing (EMNLP 2024).

**Questions:**

The proposed RPO does not include gradient descent or a policy improvement step like reinforcement learning. Does the method theoretically (even if not rigorously) guarantee convergence to an optimal policy? My intuition is that it may diverge rather than converge.

How exactly is the text gradient generation process conducted? What kinds of prompts are used for feedback generation and rewriting? Are different prompts used for different Rewriters?

Aside from being TD-style, what distinguishes this method from ORPO? As far as I understand, ORPO and MC-style RPO are essentially identical—is that correct? Why are there no comparisons with ORPO? It would help if the paper clarified which components differ from ORPO.

If I understand correctly,  V(s_t) seems to represent feedback rather than a numeric value. How is the discount factor gamma applied? Is it explicitly written in the LLM prompt as an instruction (e.g., “use this discount rate”)?

---

> ### Author Response · Authors · 2025-11-21
>
> # Additional experiments: compare with RL-based prompt editing
> > Weakness 1: Lack of comparison with RL-based prompt generation methods.
>
> Since the StablePrompt repository[1] does not specify the exact library versions used or provide the trained model weights, reproducing their multi-turn results is not feasible within the rebuttal period. Therefore, we compare against BPO[2], which trains a prompt writer using preference data based on Llama-2-7B-Chat. We run BPO with four random seeds for each system and report the mean and 95% confidence interval on the Text-to-SQL task. The experimental results is as follows.
>
> | **Method** | **GPT** | **Gemini** | **Llama-4** | **Llama-8B** | **Llama-70B** | **Mean** | **Δ%** |
> |-----------|---------|------------|-------------|--------------|----------------|---------|-------|
> | Baseline_Sharded | 0.402 ± 0.002 | 0.514 ± 0.027 | 0.206 ± 0.013 | 0.224 ± 0.016 | 0.318 ± 0.014 | 0.333 | – |
> | BPO | 0.318 ± 0.061 | 0.397 ± 0.069 | 0.220 ± 0.026 | 0.185 ± 0.028 | 0.241 ± 0.061 | 0.272 | -15.7 |
> | **RPO_TD+replay (ours)** | **0.528 ± 0.011** | **0.607 ± 0.018** | **0.383 ± 0.013** | **0.467 ± 0.022** | **0.402 ± 0.012** | **0.477** | **54.2** |
> | Oracle_Full | 0.893 ± 0.007 | 0.841 ± 0.007 | 0.729 ± 0.017 | 0.505 ± 0.017 | 0.748 ± 0.014 | 0.743 | 140.2 |
>
> The prompts optimised by BPO do not improve performance; in some cases, they even degrade it, especially when applied to systems built on different model families, e.g. GPT-4o mini or Gemini-2.0-flash.
> It is not surprising that BPO is inferior since it is optimised for revising prompts for single-turn tasks and does not generalise well to multi-turn scenarios. Its generalisability is further limited by its dependence on the Llama family as the backend, resulting in suggestions that may not transfer to GPT- or Gemini-based system agents.
>
> The full Table 1 is also shown in our revision and in the section of our response to the concern of stability.
>
> [1] StablePrompt: Automatic Prompt Tuning using Reinforcement Learning for Large Language Models. (Kwon et al., EMNLP 2024).
>
> [2] Black-Box Prompt Optimization: Aligning Large Language Models without Model Training (Cheng et al., ACL 2024)

---

> ### Author Response · Authors · 2025-11-21
>
> # API cost estimation
> > Weakness 2: While the authors claim efficiency, they do not provide quantitative evidence showing how many more API calls or tokens RPO consumes compared to APO or GPO.
>
> The token usage for feedback generation is summarised below:
> | Method | Token Count (Mean ± 95% CI) |
> |--------|---------------------------|
> | APO    | $1423.9 \pm 45.5$         |
> | GPO    | $153.0 \pm 29.4$          |
> | RPO    | $1522.58 \pm 137.2$       |
>
> We agree that textual feedback methods (APO and our proposed RPO) consume more tokens than GPO, which is a trade-off for a more informative optimisation signal.

---

> ### Author Response · Authors · 2025-11-22
>
> # Statistical reliability
> > Weakness 3: Reporting standard deviations in Tables 1 and 2 would help alleviate concerns regarding reliability.
>
> We added the 95% confidence interval in Table 1 and Table 2 here and in the revision.
>
> Table 1: different LLMs for the system agent
> | **Method** | **GPT** | **Gemini** | **Llama-4** | **Llama-8B** | **Llama-70B** | **Mean** | **Δ%** |
> |-----------|---------|------------|-------------|--------------|----------------|---------|-------|
> | Baseline_Sharded | 0.402 ± 0.002 | 0.514 ± 0.027 | 0.206 ± 0.013 | 0.224 ± 0.016 | 0.318 ± 0.014 | 0.333 | – |
> | BPO | 0.318 ± 0.061 | 0.397 ± 0.069 | 0.220 ± 0.026 | 0.185 ± 0.028 | 0.241 ± 0.061 | 0.272 | -15.7 |
> | APO | 0.374 ± 0.009 | 0.523 ± 0.008 | 0.318 ± 0.012 | 0.290 ± 0.013 | 0.336 ± 0.004 | 0.368 | 16.9 |
> | GPO | 0.458 ± 0.008 | 0.523 ± 0.016 | 0.299 ± 0.017 | 0.290 ± 0.015 | 0.308 ± 0.011 | 0.376 | 17.5 |
> | MC-style | 0.459 ± 0.018 | 0.551 ± 0.015 | 0.250 ± 0.013 | 0.346 ± 0.021 | 0.332 ± 0.007 | 0.388 | 20.4 |
> | **RPO_TD (ours)** | 0.439 ± 0.011 | 0.561 ± 0.013 | 0.336 ± 0.014 | 0.318 ± 0.009 | 0.383 ± 0.011 | 0.408 | 28.9 |
> | **RPO_TD+replay (ours)** | **0.528 ± 0.011** | **0.607 ± 0.018** | **0.383 ± 0.013** | **0.467 ± 0.022** | **0.402 ± 0.012** | **0.477** | **54.2** |
> | Oracle_Full | 0.893 ± 0.007 | 0.841 ± 0.007 | 0.729 ± 0.017 | 0.505 ± 0.017 | 0.748 ± 0.014 | 0.743 | 140.2 |
>
> Table 2: different LLMs for meta-prompting agent
> | **Method** | **GPT** | **Gemini** | **Llama-4** | **Llama-8B** | **Llama-70B** | **Mean** | **Δ%** |
> |-----------|---------|------------|-------------|--------------|----------------|---------|-------|
> | APO | 0.540 ± 0.030 | 0.560 ± 0.018 | 0.540 ± 0.018 | 0.560 ± 0.018 | 0.560 ± 0.018 | 0.552 | 31.4 |
> | GPO | 0.579 ± 0.021 | 0.541 ± 0.008 | 0.571 ± 0.046 | 0.554 ± 0.017 | 0.526 ± 0.053 | 0.554 | 32.0 |
> | MC-style | 0.567 ± 0.042 | 0.549 ± 0.010 | 0.575 ± 0.038 | 0.560 ± 0.029 | 0.572 ± 0.044 | 0.565 | 34.4 |
> | **RPO_TD (ours)** | 0.578 ± 0.037 | 0.562 ± 0.036 | 0.586 ± 0.021 | 0.594 ± 0.013 | 0.556 ± 0.042 | 0.575 | 37.0 |
> | **RPO_TD+replay (ours)** | **0.625 ± 0.038** | **0.622 ± 0.018** | **0.618 ± 0.018** | **0.622 ± 0.007** | 0.606 ± 0.020 | **0.619** | **47.3** |

---

> ### Author Response · Authors · 2025-11-22
>
> # Ablation study of feedbacker and rewriter
> > Weakness 4a: Although the paper highlights the importance of the Feedbacker and Rewriter, no experiments are conducted to replace or analyse them individually.
>
> First of all, it is not possible to isolate the feedbacker and the rewriter, since the feedbacker generates the direction of optimisation and the rewriter applies these textual gradients on the system instruction. If one module is missing, the other is not able to work. Here we briefly summarise the ablation study we have done in our paper:
> * The choice of LLMs for meta-prompting agents:
>
>   As shown in Table 2 in Section 5.1, there is *no* statistically significant difference in performance for our proposed method $RPO_{TD+replay}$ between the different choices of the LLMs for the meta-prompting agent. This demonstrates the robustness and generalisability of our proposed method.
>
> * Different settings of the rewriter:
>
>   The experience replay rewriter can reach a *better performance* as shown in Table 1 and Table 2, and can provide a *more stable training curve* as shown in Figure 4a in Section 5.2 and Figure 6 in Appendix B.
>
> * Different feedbacker algorithm: numerical only (GPO), MC-style, TD-style
>
>   TD-style achieves better performance than GPO on average (Table 1 and Table 2), and learns faster than MC-style (Figure 4a and 8). After being equipped with an experience replay rewriter, $RPO_{TD+replay}$ can achieve the best performance.
>
> * Incomplete info for feedbacker:
>
>   In Section 5.2 (Figure 4b), we already demonstrated RPO's *robustness to lower quality of feedback* signal by providing incomplete information to the feedbacker. By removing specific information, e.g. removing the user goal in the *believe* setting, removing the API call in the *subjective* setting, or removing both the user goal and API call in the *basic* setting, RPO can still improve the system, and the full setting can reach the optimal performance.
>
> * *NEW* Noisy signal from external evaluator:
>
>   We have conducted a *new* experiment on robustness to **noisy external evaluation signal**, where we flipped 20%, 40%, and 60% of the external evaluation signal (please see Figure 5 in Section 5.2 in our revision). The experiment is conducted with Gemini-2.0-flash on task-oriented dialogue, and the result is as follows.
>
>   | Noise level | Success |
>   | :--- | :---: |
>   | **clean** | $0.622 \pm 0.018$ |
>   | **20%** | $0.581 \pm 0.021$ |
>   | **40%** | $0.579 \pm 0.018$ |
>   | **60%** | $0.549 \pm 0.038$ |
>
>   The performance is robust when the noise level is up to 40%. The optimisation ability of RPO declines when the noise reaches $60\%$, which reverses the reward signal on average and naturally harms any optimisation. On the other hand, if the noise level is less than 50%, its performance is close to the clean setting. In other words, the turn-level feedback, which is not conditioned on the external feedback, provides a learning signal that, to some extent, mitigates the noise in the external evaluation signal.
>
> > Weakness 4b: The paper provides insufficient explanation of the prompts used for generating text gradients.
>
> In Section 3.1, we describe our prompt design, including estimation of a short-term reward (next turn user emotion) and long-term value estimation (future task success), inspired by TD-error. Detail of the prompt is shown in Figure 10, and one example is provided in Figure 14.

---

> ### Author Response · Authors · 2025-11-22
>
> # Guarantee convergence to an optimal policy?
> > Question 1: The proposed RPO does not include gradient descent or a policy improvement step like reinforcement learning. Does the method theoretically (even if not rigorously) guarantee convergence to an optimal policy? My intuition is that it may diverge rather than converge.
>
> Thank you for raising the concern of convergence and stability.
> First of all, even numerical RL algorithms *do not* guarantee convergence; only very few algorithms can do which usually require strong assumptions, e.g., tabular setting or full exploration.
> Secondly, we apply the concept of *experience replay* in the rewriter, which improves the stability, i.e. the blue lines, our proposed $TD-style+replay$, are more stable than the others, $TD-style$ and $MC-style$, across various choices of LLM for meta-prompt agents (see Figure 4a in Section 5.2 and Figure 7 in Appendix B).
>
> # Implementational details
> > Question 2a: How exactly is the text gradient generation process conducted?
>
> In our original version,
> * **Section 3.1 Feedback Generation**: describes how to generate the textual TD-style feedback
> * **Section 3.2 Applying Feedback to the Prompt**: details how to apply the textual gradient on the system prompt with or without experience replay
>
> Here we would like to briefly summarise the process of our proposed method.
> The feedbacker is prompted to assess user satisfaction after each system response, predict future task success, and provide suggestions for improvement, along with supporting reasons. These turn-level feedbacks are then summarised by the feedbacker, conditioned on an external evaluation signal. Finally, all dialogue-level feedback within the current iteration are consolidated into a single final feedback signal.
>
> We also added the pseudo-code of our proposed method in our revision (see Algorithm 1 in Appendix D).
> > Question 2b: What kinds of prompts are used for feedback generation and rewriting?
>
> The prompts of each LLM-based module are listed in Appendix C, including the prompt of feedback generation (MC style: Figure 10, TD-style Figure 11) and rewriter (basic: Figure 8, with experience replay: Figure 9).
>
> The prompt of the feedbacker follows the concept of ReAct [1], i.e. the feedbacker will first generate reasons, then its prediction about user satisfaction, future task success, and generate the suggestions.
> On the other hand, the prompt of the rewriter is the basic input-output style, to facilitate parsing the updated system prompt based on the response of the rewriter.
>
> [1] ReAct: Synergizing Reasoning and Acting in Language Models (Yao et al., ICLR 2023)
>
> > Question 2c: Are different prompts used for different Rewriters?
>
> The prompt of the rewriter is the same across different tasks to keep the generalisability of the proposed method.
>
>
> > Question 4: If I understand correctly, V(s_t) seems to represent feedback rather than a numeric value. How is the discount factor gamma applied?
>
> Since the task is episodic, you can consider that the discount factor is set to one [2], so there is no need to specifically take care of it in the text. We clarified this in our revision.
>
> [2] Sutton et al., Reinforcement Learning: An Introduction. 2nd ed. The MIT Press, 2018.
>
> # Baseline selection justification
> > Question 3: Aside from being TD-style, what distinguishes this method from ORPO? As far as I understand, ORPO and MC-style RPO are essentially identical—is that correct? Why are there no comparisons with ORPO? It would help if the paper clarified which components differ from ORPO
>
> We agree that ORPO [3] is related to one of our major baselines GPO [4]. ORPO uses *numerical scores* from an objective function evaluator and conditions the rewriter on paired (solution, score) inputs. GPO builds on this idea but introduces a more fine-grained mechanism, which has been shown to outperform ORPO across a range of tasks. This is why we focus on GPO as the stronger and more relevant numerical-feedback baseline.
>
> On the other hand, another baseline MC-style (referred to TextGrad in [5]) is different to ORPO, leveraging textual feedback from an LLM agent instead of numerical scores from an objective function. In comparison to our proposed method RPO, we explore multi-turn tasks, investigating which kind of information is essential for prompt optimisation and how robustness of our method to various model choice (Table 2 in Section 5.1), incomplete information for feedbacker (Figure 4b in Section 5.2), and noisy numerical scores from objective function (our new experiment, see Figure 5 in Section 5.2). Please check our revision for more details.
>
> [3] Large Language Models as Optimizers (Yang et al., ICLR 2024)
>
> [4] Unleashing the potential of large language models as prompt optimizers: Analogical analysis with gradient-based model optimizers (Tang et al., AAAI 2025)
>
> [5] Optimizing generative AI by backpropagating language model feedback (Yuksekgonul et al., Nature 2025)

---

### Official Review · Reviewer_foqG · 2025-11-02

**Soundness:** 2
**Presentation:** 3
**Contribution:** 2
**Rating:** 2
**Confidence:** 4

**Summary:**

This paper describes a prompt optimization method for improving multi-turn interactive tasks with LLMs. Unlike prior work which focuses on single-turn prompt optimization or self-feedback of outputs, the authors propose a framework called Reinforced Prompt Optimization (RPO) that treats the instruction prompt itself as a parameter to be refined via turn-level feedback and prompt rewriting. The model interacts over multiple turns in tasks such as text-to-SQL, task-oriented dialogue and medical QA; a “feedbacker” LLM or human expert gives incremental feedback and a “rewriter” LLM updates the prompt to improve long-term planning. Experiments on three multi-turn tasks show that RPO improves success rates and generalizes across different LLM base models.

**Strengths:**

1. The idea of treating the instruction prompt as a dynamic, optimizable parameter in multi‐turn settings offers a different angle to the other approaches that require tuning of model weights.
2. The experimental setup is broad: covering multiple tasks (SQL generation, dialogue, medical QA) and a mixture of open and closed source LLMs, which supports generalizability.
3. The experiments uses a clear setup with ablations, e.g. showing turn-level feedback vs full-trajectory feedback, to highlight which components contribute to improvement.

**Weaknesses:**

1. The idea of using LLM to provide feedback and adjust system prompt has been widely discussed, e.g. [1] and [2]
2. It's unclear how the choice of the base model for the feedbacker and rewritter would affect the final system performance. What's the principles or best practice for selecting the underlying models for this modules?
3. Practical aspect, it’s unclear how many epochs, how much interaction data, or how timely prompt updates can be in a live system.


[1] Self-Refine: Iterative Refinement with Self-Feedback
[2] Reflexion: Language Agents with Verbal Reinforcement Learning

**Questions:**

1. Any principle or best practice one should follow to select the underlying models for the feedbacker and the rewritter?
2. What happens if the feedbacker LLM is of lower quality or the feedback is noisy, how robust is RPO in that case?

---

> ### Author Response · Authors · 2025-11-22
>
> We are happy to see that you recognise our work as novel and highlight that unlike others we optimise the prompt in a reinforcement learning-inspired set-up, treating the prompt as a parameter that can be optimised with a long-term objective.  We also appreciate your recognition of our clear ablation studies that highlight the contribution of each component in the proposed framework. Here is our reply to your concerns and questions:
>
> # The Difference between the Proposed RPO and Self-Refine Methods
> > Weaknesses 1: The idea of using LLM to provide feedback and adjust system prompts has been widely discussed
> We are a bit surprised by this comment in particular, in the view that you recognise that our approach provides a different angle for model optimisation in *Strength 1*.
> Nevertheless, we will briefly comment on the difference between Self-refine/Reflexion to our work:
>  * RPO (ours): Focus on optimising the **system instruction** during **training**,
>  * Self-refine/Reflexion: Polish the **system's output** per **every inference call**.
>
> By optimising the prompt, RPO can reduce the serving cost of the generate-feedback-rewrite cycle during deployment. In addition, these two approaches are not mutually exclusive; for example, RPO could be used to improve the prompts of the Reﬂexion or Reﬁner agents in the self-reﬁne pipeline, which is an interesting future work.
>
> We already cited Reflexion in the related section in the current version and will add Self-Refine in our revision. Thank you for suggesting this relevant work.

---

> ### Author Response · Authors · 2025-11-22
>
> # The Choice of LLMs for Meta-Prompting Agents
> > Weakness 2 & Question 1: It's unclear how the choice of the base model for the feedbacker and rewritter would affect the final system performance. What are the principles or best practices for selecting the underlying models for these modules?
>
> You might have missed this, but based on our experiment in the second part of Section 5.1 (Table 2), we show that the impact of the choice of base model for the feedbacker and rewriter is not critical for RPO, i.e. there is no statistical significant difference in performance for our proposed method $RPO_{TD+replay}$ between the different choices of the LLMs for the meta-prompting agent. This demonstrates the robustness and generalisability of our proposed method.
>
> # Practical Aspects and Hyperparameter Choice
> > Weakness 3: Practical aspect, it’s unclear how many epochs, how much interaction data, or how timely prompt updates can be in a live system.
>
> Thank you for raising the practical issue. The optimal hyperparameters (number of epochs or data amount) naturally differ based on the task complexity, similar to other optimisation algorithms.
> * Epochs and convergence: According to Figure 4a and Figure 6, the training curve for task-oriented dialogue converges around epoch 3. This suggests that a minimum optimisation of 3 epochs is a recommended setting, which is also aligned with related works, GPO[1] and TextGrad[2], where they also set the epoch number from 3 to 5.
> * Number of training trajectories: As mentioned in Section 4.3, 10 interactions are used for optimisation.
> * Timeliness: We conduct a human-in-the-loop experiment as mentioned in Sections 4.1 and 5.3 to demonstrate the feasibility and timeliness of our proposed method. As shown in the following table, generating feedback for one dialogue is less than 10 seconds on average.
>
> Here is a summary of the hyperparameters from our experiments, where the estimated time to generate the feedback per trajectory is based on gemini-2.0-flash.
>
> | Feature | Text-to-SQL | Task-oriented dialogue | Medical Question-answering |
> | :--- | :---: | :---: | :---: |
> | **\# of training Epochs** | 5 | 8 | 3 |
> | **\# of training trajectories per epoch** | 10 | 10 | 10 |
> | **time of feedback generation per trajectory** | $5.91 \pm 0.09$ | $6.94 \pm 0.07$ | - |
>
>
> [1] Unleashing the potential of large language models as prompt optimizers: Analogical analysis with gradient-based model optimizers (Tang et al., AAAI 2025)
>
> [2] Optimizing generative AI by backpropagating language model feedback (Yuksekgonul et al., Nature 2025)
>
> We have included the hyperparameter details in the updated version.

---

> ### Author Response · Authors · 2025-11-22
>
> # Impact of the Quality of Feedbacker
> > Question 2: What happens if the feedback LLM is of lower quality or the feedback is noisy? How robust is RPO in that case?
>
> This is an excellent point, thank you for raising this. In Section 5.2 (Figure 4b), we already demonstrated RPO's robustness to incomplete information in the feedbacker input. By removing specific information, e.g. removing the user goal in the *believe* setting, removing the API call in the *subjective* setting, or removing both the user goal and API call in the *basic* setting, RPO can still improve the system with these suboptimal feedback signals, and the full setting can reach the optimal performance.
>
> To substantiate this more, we have conducted a new experiment on robustness to noisy external evaluation signal, where we flipped 20%, 40%, and 60% of the external evaluation signal (please see Figure 5 in Section 5.2 in our revision). The experiment is conducted with Gemini-2.0-flash on task-oriented dialogue, and the result is as follows.
>
> | Noise level | Success |
> | :--- | :---: |
> | **clean** | $0.622 \pm 0.018$ |
> | **20%** | $0.581 \pm 0.021$ |
> | **40%** | $0.579 \pm 0.018$ |
> | **60%** | $0.549 \pm 0.038$ |
>
> The performance is robust when the noise level is up to 40%. The optimisation ability of RPO declines when the noise reaches $60\%$, which reverses the reward signal on average and naturally harms any optimisation. On the other hand, if the noise level is less than 50%, its performance is close to the clean setting. In other words, the turn-level feedback, which is not conditioned on the external feedback, provides a learning signal that, to some extent, mitigates the noise in the external evaluation signal.

---

### Author Response · Authors · 2025-12-02
**Summary comment**

Dear PC, AC, and reviewers,

Thank you for the effort you put into reviewing our manuscript. Thank you to the reviewers for the time they invested in reading the paper and making suggestions that improved our submission.

Here we would like to summarise the discussion during the rebuttal as follows:

We appreciate that the reviewers emphasise the strengths of our work:

* **Novelty**: Treating prompts as optimisable parameters offers a new angle on model optimisation and directly addresses LLMs’ suboptimal performance in multi-turn tasks (Reviewers foqG, qCy2).

* **Robustness and Generalisation**: RPO is extensively evaluated across diverse tasks (Text-to-SQL, Task-oriented Dialogue, and Medical Question-answering) and both open- and closed-source LLMs, with clear ablations that support our claims (Reviewers foqG, n39h, qCy2).

* **Practicality and Efficiency**: The method is simple, elegant, and easy for researchers to apply to their own tasks (Reviewers n39h, mY4n), and is more cost-effective than existing prompt-optimisation and gradient-based approaches (Reviewer mY4n)

* **Solid Theoretical Grounding**: The TD-style fine-grained feedback formulation is well-motivated and well-suited for multi-turn optimisation (Reviewer mY4n).

All reviewers also noted that the paper is clearly written and easy to follow.

In addition, we summarise how we addressed the reviewers’ concerns:

* **Robustness to Low-Quality Feedback** (Reviewers foqG, qCy2, mY4n):

  Section 5.2 (Fig. 4b) already shows that RPO remains effective with feedback generated with incomplete turn-level information, e.g., missing user goals or API calls.

  To further validate robustness, we added a **new experiment** on noisy external evaluation signals, flipping 20%, 40%, and 60% of the reward labels.
  RPO is stable up to 40% noise, and degrades in a large margin when noise exceeds 50%, where the reward becomes inverted on average.
  This demonstrates that RPO’s turn-level feedback mitigates noise in the external signal to some degree.

  In summary, RPO is robust to noisy or incomplete reward signals, though cleaner feedback naturally yields stronger optimisation.

* **Missing Comparison with RL-Based Prompt Optimisation** (Reviewer n39h):

  We added a comparison with Black-Box Prompt Optimization (BPO) [1] (Section 5.1, Table 1).
  BPO fails to improve performance, and particularly degrades it when applied to models different to the Llama family (e.g., GPT-4o mini, Gemini-2.0-flash). This is expected since BPO is designed for single-turn prompt editing and does not generalise to multi-turn settings.

[1] Black-Box Prompt Optimization: Aligning Large Language Models without Model Training (Cheng et al., ACL 2024)

* **Statistical Reliability** (Reviewer n39h):

  We added 95% confidence intervals to Table 1 and Table 2 in the revised manuscript.

* **Qualitative Analysis** (Reviewer mY4n):

  Appendix C now includes examples of turn-level, dialogue-level, and final feedback (Figures 8–10), as well as comparisons with MC-style feedback and the resulting optimised prompts (Figures 12–14), illustrating why TD-style feedback leads to greater improvements.

* Practical Considerations
  * Impact of *model choice* (Reviewers foqG, n39h): Table 2 (Section 5.1) shows no statistically significant performance differences across different LLMs used as feedbackers/rewriters, confirming RPO’s robustness to model choice.

  * Hyperparameter settings (Reviewers foqG, n39h, mY4n): We clarified choices for epochs, number of trajectories, and meta-prompting prompts, and added pseudocode in Appendix E.

  * Cost analysis (Reviewers foqG, n39h): We added analyses of runtime and token usage, confirming the method’s efficiency.

We addressed **all** reviewers' concerns, and Reviewers qCy2 and mY4n explicitly noted that our responses and new experiments resolved their issues.

Finally, we would like to draw AC's attention to the positive and, in parts, very positive formulation of the reviews. At the same time, we feel that the scores from Reviewers foqG and n39h do not adequately reflect the narrative of their reviews. In particular, we are a bit puzzled with the review (foqG), where on one hand the reviewer says that this is a new angle on this widely researched problem (strengths) and that this is a widely researched problem (weaknesses). It is not clear to us how the same claim can be part of both weaknesses and strengths, but when one takes into account everything that the reviewers wrote, this point should be considered a strength.

---

### Meta-Review · Area_Chair_PeHp · 2026-01-07

**Summary:**

This paper introduces RPO, a meta-prompting framework for improving long-term planning in multi-turn LLM interactions by updating only the task instruction prompt rather than model weights. The authors propose generating turn-by-turn natural-language feedback (inspired by TD-style signals) and using an experience-replay-like prompt rewriter to iteratively refine the task instruction. This paper reports improvements on multi-turn tasks such as text-to-SQL and task-oriented dialogue. However, as pointed by reviewers, weaknesses include a weak evidenced RL connection. Although the paper frames TD-style feedback as short-term reward (next-turn user emotion) and long-term value (future task success), the approach remains iterative prompt rewriting driven by LLM-generated feedback. AC thinks that the statement that the method is inspired by RL is not very convincing and objective. AC encourages the authors to tune down the statement. Further, while the paper studies feedback styles (MC-style vs TD-style) and the inputs provided to the feedbacker, these experiments do not isolate whether the reward fields (emotion prediction and success forecasting) are the key drivers, as pointed out by a reviewer. Because the paper does not provide controlled, mechanism-revealing ablations, It is unclear whether the gains come from the proposed RL-style credit assignment, or from generic iterative prompt refinement. AC would suggest that doing some ablations at least, such as setting the feedbacker result as null, not sure if this would work out eventually but it would good to try some ideas.

The scores of this paper are 2,2,6,6. R4 mentioned they would increase the score (not sure what their original rating was). R3 said "thank all authors for answering" which looks neutral.

The AC recommends reject after careful calibration. The rationale is given above. Authors are encouraged to update their manuscript accordingly.

**Reviewer Concerns:**

See above.

**Reviewer Scores:**

see above.

---

### Decision · Program_Chairs · 2026-01-26

Reject